# Luminescence dating approaches to reconstruct the formation of plaggic anthrosols

Jungyu Choi[1,2], Roy van Beek[1,3], Elizabeth Chamberlain[1,2], Tony Reimann[4], Harm Smeenge[5], Annika van Oorschot[1], Jakob Wallinga[1,2]

[1] Soil Geography and Landscape Group, Environmental Sciences Group, Wageningen University, Wageningen, 6700AA, the Netherlands
[2] Netherlands Centre for Luminescence Dating, Wageningen University, Wageningen, 6700AA, the Netherlands
[3] Cultural Geography Group, Environmental Sciences Group, Wageningen University, Wageningen, 6700AA, the Netherlands
[4] Institutes of Geography, University of Cologne, Cologne, 50674, Germany
[5] Bosgroepen, Ede, 6717LL, the Netherlands

*Correspondence to*: Jungyu Choi (jungyu.choi@wur.nl)

**Abstract.** Plaggic anthrosols demonstrate the significant and widespread influence of agriculture on the landscape of northern Europe and testify to increased land-use intensity over the last millennium. However, a lack of established chronologic methods to interrogate these soils has hindered research on their formation history, so the timing and process of plaggic anthrosol development remain poorly quantified. Recently, luminescence dating methods have emerged as a tool for tracing the past movement of grains, including within the soil column. This study combines two primary luminescence methods -- single-grain feldspar infrared (IRSL) and post-infrared infrared (pIRIR) measurements, and small-aliquot (or, multigrain) quartz optically stimulated luminescence (OSL) -- to reconstruct the formation of a plaggic anthrosol at Braakmankamp (eastern Netherlands). Toward this aim, we present a new method to identify well-bleached single grains of feldspar using the ratio of the grains' IRSL and pIRIR signals as a 'filter'. The results provide both methodological and applied archaeological insights. Both small-aliquot quartz OSL and single-grain feldspar pIRIR ages yield reliable ages for plaggen deposits when the new filtering approach is used to remove poorly bleached feldspar grains from the analysis. Single-grain pIRIR feldspar has the added benefit of revealing complex soil formation histories for naturally bioturbated deposits including those at the base of the plaggen layer. Augmenting this information with conventional quartz OSL dating builds confidence in the geochronologic record and allows us to reconstruct the timing and processes of plaggic anthrosol formation in Braakmankamp. According to the luminescence dating results, land clearance occurred around 900–1000 years ago, and accumulation of plaggen material began around 700–800 years ago. The average accumulation rate of plaggen material is estimated at ~ 1.1 mm/yr.

## 1 Introduction

Human agricultural activities are highlighted as one of the primary factors of human landscape alteration by several researchers (Briggs et al., 2006; Delcourt et al., 2004; Denevan, 1992). The European landscape has been substantially transformed by

agricultural activity since the settling of the first farmers during the mid-Holocene (Kaplan et al., 2009). In this research, we focus on the highly engineered Dutch landscape, where early agricultural activities date to at least 4300 BCE (Huisman and Raemaekers, 2014). Pre-modern agricultural practices in the Netherlands can still be observed in the remains of prehistoric

field systems (previously often described as Celtic fields, see Arnoldussen, 2018) and (early) historical open fields characterized by plaggic anthrosols. The creation of anthrosols is one of the key elements to understanding the landscape alteration caused by agricultural practices.

Plaggic anthrosols (Dutch: *plaggendekken*) are anthropogenic soils that have resulted from fertilizing nutrient-poor sandy soils. They develop through the artificial raising of the fields by the continuous annual input of sods (Dutch: *plaggen*), often mixed

with manure. The sods came from various sources, which include heathlands and valleys (Groenman-van Waateringe, 1992; Pape, 1970). Plaggic anthrosols are commonly found in the sandy areas of North-West Europe, including the Netherlands, Belgium, northern Germany, and Denmark (Blume and Leinweber, 2004; Giani et al., 2014; Pape, 1970; Spek, 2004). In the Netherlands, the most typical type of plaggic anthrosols overlies a layer of xeropodzol soils, or hydromorphic sandy soils (Pape, 1970), but this may differ according to spatiotemporal circumstances.

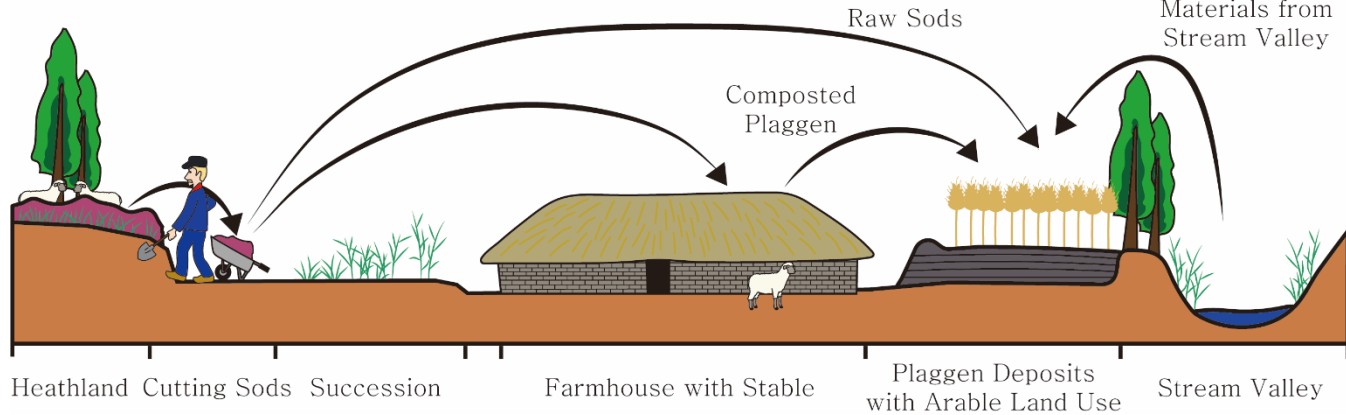

**Figure 1: Schematic overview of the method of plaggen agriculture. Modified after the picture by Klaus Thierer presented in Giani et al. (2014) using observations from our study site, at Braakmankamp. Typical plaggen agriculture involves the mixing of sods with animal manure, but there are cases where raw sods are applied to the arable fields as well (Smeenge, 2020). The sods are typically collected from heathlands, but materials from the adjacent stream valley were also used at Braakmankamp (Smeenge, 2020).**

The introduction of plaggen agriculture was one of the major elements of land-use intensification in the Middle Ages (Ellis et al., 2013; van Mourik et al., 2012). The continuous annual input of plaggen enabled yearly cultivation of a field plot without fallowing periods (McKey, 2021). The formation of plaggic anthrosols had an impact beyond the arable fields, since the plaggen management activities involved areas surrounding the villages and arable fields, such as heathlands and valleys. In such areas, the difference in elevation, vegetation, and groundwater level was significantly altered by activities related to

plaggen agriculture (Blume and Leinweber, 2004). Despite the significant influence of plaggen agriculture on the land-use intensification of the Netherlands in the Middle Ages, relatively little is known about the degree and timing of its impact. Although there has been an estimation of how land-use intensity increased with the introduction of plaggen agriculture on a

regional level (Blume and Leinweber, 2004), a quantitative estimation of the land-use intensification from the soil itself is absent. Furthermore, the formation history of plaggic anthrosols is poorly understood, mainly due to the inconsistency between

different sources of dating, such as historical data, archaeological records, radiocarbon dates, pollen analyses, and luminescence dating (Bokhorst et al., 2005; Giani et al., 2014; van Mourik et al., 2012; van Mourik et al., 2011).

Luminescence approaches, which have emerged since the 1980s for dating sediments (e.g., Huntley et al., 1985), may offer a solution to resolve the formation history of plaggen soils. This radiometric dating method makes use of energy that accumulates in the crystal lattice of quartz or feldspar minerals. The energy is stored when the minerals are removed from light (buried)

and it is released as a measurable flux of light upon exposure to heat or (sun)light in a process known as 'bleaching' (or, 'zeroing', 'resetting'). Both quartz and feldspar offer unique advantages and disadvantages that are detailed in Section 2. Quartz is conventionally used for dating when feasible because it yields a stable and readily bleached signal (e.g., Godfrey-Smith et al., 1988). Yet, feldspar has high potential as a sediment tracer (e.g., Wallinga et al., 2019) because it is more readily measured on a single-grain level than quartz and it can provide multiple signals with different bleachability from the same

grain (Li et al., 2014; Thomsen et al., 2008).

In this research, we explore the utility of single-grain feldspar luminescence to plaggic anthrosols to: 1) establish the timing of the initial stages of plaggen agriculture, and 2) identify changes in soil-mixing intensity during the evolution of the plaggic anthrosols. To examine whether single-grain feldspar methods can contribute to investigating the timing and process of the formation of plaggic anthrosols we address three research questions: 1) How can well-bleached grains be identified for feldspar

single-grain pIRIR dating?; 2) Do results from feldspar single-grain pIRIR dating agree with more conventional small-aliquot quartz optically stimulated luminescence (OSL) dating?; 3) What new information on the evolution of plaggic anthrosols is gained from combining quartz OSL and feldspar single-grain pIRIR analyses? We focus on a high-resolution record of a single site with a plaggic anthrosol to answer these questions. The methods developed and results obtained have broad application potential for dating and reconstructing soil formation processes in human-influenced landscapes of the world.

**2 Dating plaggic anthrosols**

It is generally accepted that during the Middle Ages, open-field agriculture was adopted throughout the northern half of the European landscape (Taylor, 1981; Renes, 2010). Since the sandy soils in coversand landscapes were poor in nutrients, agricultural activities were mostly limited to areas with specific geomorphological and hydrological characteristics. These tended to be relatively high and large coversand ridges, often bordered by valleys (Deeben et al., 2007; Renes, 2018; Spek,

2004). In the Dutch landscape, the open-field system was adopted in the form of so-called *essen* (singular: *es*). *Essen* are large and open arable fields, which are often (partially) communally used. They are generally parceled, based on the property rights of different parts of the arable field (Deeben et al., 2007).

In arable areas like *essen* and other types of open fields, the fertility of the soils was maintained or improved by the application of plaggen agriculture (Edelman, 1950; Pape, 1970; Renes, 2018; Spek, 2004). The exact timing of the emergence of plaggic

anthrosols is still under debate, due to a scarcity of information and the general incompatibility of different types of data (Dyer et al., 2018; Renes, 2010; Taylor, 1981). A number of studies place the substantial spread of plaggen agriculture in northwestern Europe at the beginning of the High Middle Ages, around 1000 CE (Blume and Leinweber, 2004; Giani et al., 2014), based on various methods. The applied methods range from the '1 mm per year theory', in which the thickness of the plaggen is divided by an assumed accumulation rate of 1 mm per year, to isotopic dating methods (Spek, 2004). Early radiocarbon dating attempts in the 1960s produced dates as early as 600 to 800 CE (Giani et al., 2014). In one case, a Dutch plaggic anthrosol was dated between 500 BCE to 100 CE based on pollen analysis (Blume and Leinweber, 2004; Giani et al., 2004). The oldest plaggic anthrosols that have been reported are located on the German north-coast islands of Sylt and Föhr, which are assumed to be more than 3000 years old (Blume and Leinweber, 2004). Given the widespread occurrence of plaggic anthrosols in the northern part of Europe, plaggen agriculture may have been practiced in different regions at different times. Improving dating methods for plaggic anthrosols would be an important contribution to the understanding of the temporal position of open-field systems.

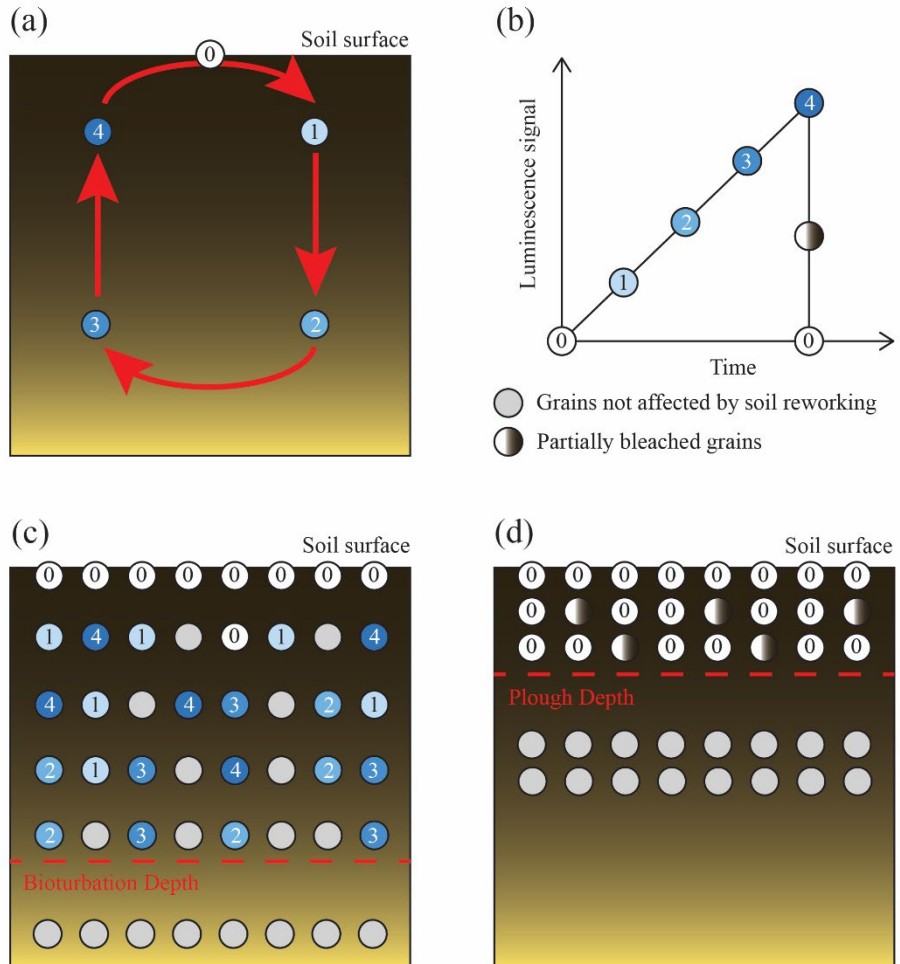

**Figure 2: Conceptual diagrams illustrating the expected luminescence signals of grains experiencing soil reworking by agricultural activities and bioturbation, which (a) deliver grains from the surface to depth and from depth to the surface over time. (b) Luminescence signals accumulate with time while the grains are buried and are bleached by light exposure when the grains are at the surface position. While some grains may be fully bleached, others may be partially bleached if they are not sufficiently light-exposed at the soil surface. (c) Exposed scenario when grains are reworked by bioturbation, and (d) expected scenario when grains are reworked by agricultural activities, mainly ploughing. The soil reworking process by bioturbation is relatively slow compared to ploughing, so bioturbated grains may spend more time at the surface than sediments. Bioturbation is also less intense and can reach a greater depth than ploughing. This may result in greater heterogeneity in the equivalent doses of well-bleached bioturbated grains at a given depth. In contrast, ploughed grains are more likely homogenised by the intense mixing process and exhibit less heterogeneity in the equivalent doses of the well-bleached grains.**

A number of more recent publications dated the soils using optically stimulated luminescence (OSL) (Bokhorst et al., 2005; van Mourik et al., 2011; van Mourik et al., 2012). Luminescence dating is advantageous for sedimentary landscapes because the obtained ages reflect the moment of the depositional event, provided that the material is exposed to light prior to deposition and shielded from light since that time and that a suitable protocol is used. The sand grains within agricultural soils like plaggic anthrosols may have varied individual histories due to the combined processes of natural bioturbation, ploughing, and accretion

(e.g., through sodding of the plaggen) (Fig. 2). With luminescence methods, it may be possible to interrogate individual grains to parse these processes within the soil column while measuring multiple grains will yield an average burial history of the deposits. Comparison of OSL ages to radiocarbon dates of plaggic anthrosols has yielded a discrepancy (van Mourik et al., 2011; van Mourik et al., 2012), with the former providing much younger ages than the latter. The research by van Mourik et al. (2011) concluded that the luminescence ages reflect the deposition of plaggic horizons while the radiocarbon dates are indicative of the organic material related to the beginning of agricultural land use and are largely affected by the mixing of organic matters of different ages (van Mourik et al. 2011; Wallinga et al., 2019).

Although OSL dating of plaggic anthrosols has yielded consistent and seemingly reliable results for the accumulation phase, which generally begins around 1600 CE in the Netherlands (van Mourik et al., 2011; van Mourik et al., 2012), several challenges remain. Firstly, robust dating of the initial stages of the development of plaggic anthrosols has proven to be problematic as only part of the grains in the lowest parts of plaggic deposits are exposed to light. Secondly, standard methods provide little information on the intensity of mixing as a function of time. These challenges are related to within-aliquot averaging effects for small-aliquot quartz OSL dating (Cunningham et al., 2011; Wallinga, 2002).

The averaging effect of the multi-grain luminescence dating can be overcome by performing equivalent dose ($D_e$) measurements at a single-grain level. The ideal situation would be to perform a measurement on single-grain quartz since the fast-component OSL signal of quartz bleaches faster than feldspar infrared stimulated luminescence (IRSL) or pIRIR signals (Kars et al., 2014b; Murray et al., 2012). However, in sediments from the North European Plain and many cases elsewhere, the luminescence sensitivity of quartz is low, with typically less than 3–5 % of the grains providing sufficient OSL signal to determine $D_e$ values (Cunningham et al., 2015; Duller, 2008). Therefore, single-grain quartz OSL dating is not practical in most settings.

As an alternative to quartz OSL, feldspar grains can be useful for single-grain IRSL or pIRIR dating, since they are more sensitive, with approximately 50 % of the grains providing sufficiently bright signals to produce a $D_e$ (Reimann et al., 2012). However, feldspar has its own deficiencies, which hinder it from being commonly used as a natural dosimeter. The first problem is anomalous fading: a-thermal loss of the luminescence signal with time (Spooner, 1994; Wintle, 1973), causing luminescence ages to underestimate the actual burial age. To largely avoid the problem of fading, the use of pIRIR signals, measured at elevated temperature(s) following the evacuation of the lower temperature IRSL signal, was suggested by Thomsen et al. (2008). The pIRIR signals are increasingly stable (i.e., less affected by anomalous fading) at higher stimulation temperatures (Buylaert et al., 2012; Cheng et al., 2022; Kars et al., 2014b; Reimann and Tsukamoto, 2012). However, the pIRIR signals reset much more slowly upon exposure to light than feldspar IRSL or quartz OSL signals (Kars et al., 2014b; Thomsen et al. 2008). Consequently, pIRIR signals of fewer grains will be completely reset compared to other luminescence signals and pIRIR-derived ages are more likely to overestimate the burial age. pIRIR signals are increasingly hard to bleach at higher stimulation temperatures (e.g. Kars et al., 2014b).

Therefore, in this research, we investigate appropriate measurement parameters for single-grain feldspar luminescence dating of the plaggen and underlying deposits, seeking an optimal compromise between bleachability and stability. In addition, we develop a new approach to identify those grains for which the pIRIR signal is well bleached.

## 3 Materials and Methods

### 3.1 Site information and samples

#### 3.1.1 Site information

The sampling took place at a site named Braakmankamp, where ~ 1 m thick plaggic anthrosol development is present. Braakmankamp is located in the eastern Netherlands, south of Denekamp, Overijssel (Fig. 3, inset). This region is a part of the 'European Sand Belt', which extends from Northwestern Europe to Poland, and the Baltic region. The majority of the landscape is covered by aeolian coversand deposits of the last glacial (Weichselian, OIS 4-2). In the Netherlands, these aeolian sands are
160 characterized by a fairly uniform grain size, which ranges from 105 to 420 μm, and form hummocky landscapes with sand dunes varying in height and slope values (Koster, 2009). During the Holocene, and especially after 1,000 BCE, coversands were locally reactivated under the influence of increased human pressure resulting in drift sand often deposited in dunes (Pierik et al., 2018). The Braakmankamp site is located in a coversand landscape dissected by the river Dinkel and its tributaries forming 'sand islands' of different sizes. Braakmankamp positions itself on one of the sand islands on the east of the Dinkel
(see Fig. 1). It may have been one of the first reclaimed sites in this region because of its proximity to the Dinkel, and its relatively large extent, which qualifies as a favourable settlement area (Groenewoudt and Lubberink, 2007). The site has been used as an agricultural field, but has not been deep-ploughed in the past 50 years, and therefore, was selected as a suitable research site.

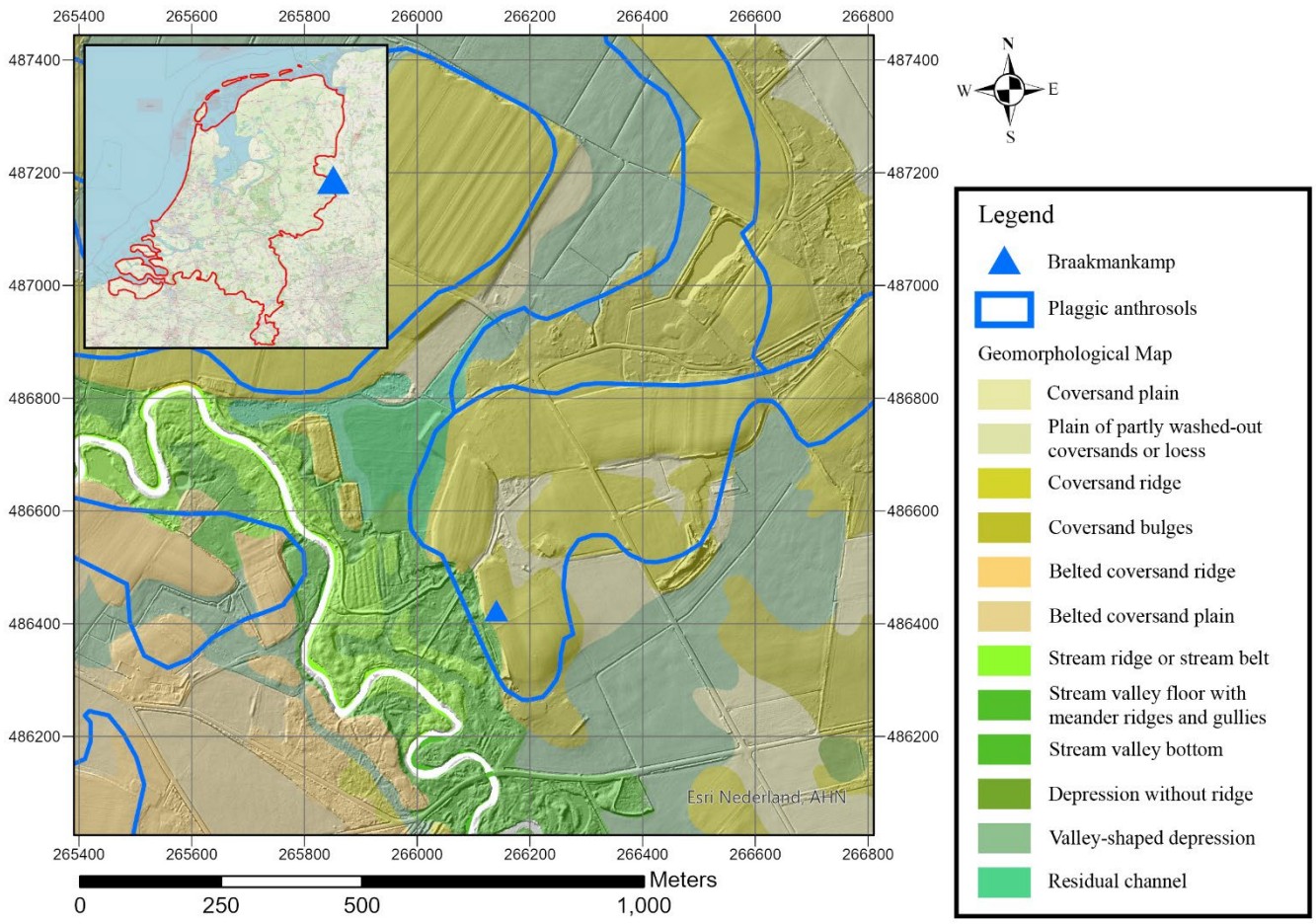

Figure 3: The location of the Braakmankamp site on the geomorphological map of the Netherlands (BRO, 2023). Plaggic anthrosols occur mostly in areas with coversand deposition. The projected coordinate system is Amersfoort / RD New (EPSG:28992).

A pit was dug at the Braakmankamp site for the documentation of the soil layers (Smeenge, 2020; van Oorschot, 2018) (Table 1), and for the collection of samples for luminescence dating and pollen analysis. The pit was located in the middle of a field plot, which is positioned on the outer border of a larger area covered with plaggic anthrosols (Fig. 3). The field plot is adjacent to a stream valley, including a tributary of the Dinkel. The soil was identified to be a plaggic anthrosol. At all depths, the texture of the soil is weakly loamy sand, with a median grain size of 210 µm. The plaggen layer is divided into 1Aap, and 2Aap, distinguished based on the colour which may reflect differences in the plaggen material used. It is probable that the brighter colour of 1Aap originated from forest/heather plaggen, and the darker colour of 2Aap is derived from plaggen containing more organic materials, which would likely have been collected from the adjacent Dinkel valley (Smeenge, 2020). Below the plaggen horizon is the non-plaggic 2Ap horizon. The 2Ap horizon has a lighter colour than the overlying Aap horizons, which is due to the inclusion of grey-coloured grains. The colour of these greyish particles resembles the colour of the eluviation horizon of podzols. Below the 2Ap horizon is the brown weathering horizon (2Bw). A coversand layer, which

is the parent material, underlies the soil matrix, and gleiing is observed in this layer (Cg). Considering the presence of the Bw layer, the soil that formed prior to the practice of plaggen deposition is a brown forest soil.

**Table 1: Description of soil horizons at the sampling location. The descriptions of the soil horizons are based on the Dutch soil classification system (de Bakker and Schelling, 1989). Translation for the description of the small-letter suffix provided by B. Makaske and G.J.W.C. Peek.**

| Depth (cm) | Horizon | Colour | Remarks | Samples | Soil Profile |
|---|---|---|---|---|---|
| 0 - 30 | 1Aap | 10YR 2/2 | Brick pieces present | NCL-1117134 (22 cm) |  |
| 30 - 50 | 1Aap/2Aap | 10YR 3/2 | Brick pieces present | NCL-1117133 (31 cm)<br>NCL-1117132 (41 cm) | |
| 50 - 105 | 2Aap | 10YR 3/3 | | NCL-1117131 (50 cm)<br>NCL-1117130 (60 cm)<br>NCL-1117129 (70 cm)<br>NCL-1117128 (81 cm)<br>NCL-1117127 (96 cm)<br>NCL-1117126 (101 cm) | |
| 105 - 120 | 2Ap | 10YR 4/2 | Grey coloured stains from podzol | NCL-1117125 (112 cm) | |
| 120 - 130 | Bw | 10YR 4/3.5 | | NCL-1117124 (123 cm) | |
| 130 - 170 | Cg | 2.5Y 5/4 | Fossil gley mottles of cm-scale | NCL-1117123 (142 cm)<br>NCL-1117122 (165 cm) | |

| **Small-letter suffix after main horizon code in the Dutch soil classification** | |
|---|---|
| a | Horizon that consists of material that was brought from else-where (e.g., a plaggen cover on an old arable field or an anthropogenic sand cover). |
| g | Horizon with rusty mottles. |
| p | Horizon that is regularly ploughed. |
| w | Code that may indicate three different phenomena:<br>1) weathering and incorporation of moder humus (in moder-podzol soils)<br>2) B and C horizons with soil aggregates (in clay or sandy clay soils)<br>3) a strongly weathered peaty C horizon |

Archaeological data demonstrates that the Twente region has been occupied by humans since the Late Palaeolithic age (van Beek et al., 2015). Large coversand ridges along the Dinkel valley have been favourable habitation sites since late prehistoric times. One of the earliest pieces of evidence of human occupation in the vicinity of the study area is found in Mekkelhorst, adjacent to Braakmankamp. In Mekkelhorst an Iron Age settlement has been identified. Since the Middle Ages, farmers formed *essen* on the sandy ridges east of the Dinkel (Smeenge, 2020). The Braakmankamp site is located on a part of such an *essen* complex. The suffix 'kamp', common in the eastern Netherlands, refers to a (generally fairly small) arable field, whereas 'Braakman' probably is an old family name.

### 3.1.2 Sampling

Thirteen luminescence dating samples were collected in a vertical sequence at the Braakmankamp site, with at least one sample from each of the identified soil horizons (Table 1). Each sample was collected by horizontally hammering in a sampling tube with a length of 20 cm and a diameter of 4.5 cm into the excavated and cleaned soil profile. After removing the sampling tubes from the soil profile, both ends of the tubes were sealed with lids and tape to prevent exposure to sunlight. For dose rate measurements, additional soil material was collected from the same depth as the sampling tubes and placed into a plastic bag.

### 3.2 Sample preparation

All sample preparations were performed under safelight conditions at the laboratory of the Netherlands Centre for Luminescence dating (NCL), at Wageningen University. Materials from the outer 3 cm of the sampling tubes, which may have been exposed to light during sampling, were used for dose rate measurements. Of the remaining material, 100 g was wet-sieved to obtain 212–250 µm sand grains. These were cleansed through magnetic separation, removing magnetic particles. After magnetic separation, the samples were treated with 10 % HCl for 1 hour to remove carbonates and 10 % H2O2 for more than 15 hours to remove organic materials. Subsequently, the samples went through a density separation process using LST heavy liquid of 2.58 g/cm$^3$, separating K-feldspars (2.57 g/cm$^3$), and quartz (2.64 g/cm$^3$) grains. The heavy fraction (2.58 g/cm$^3$) was etched with HF for 40 minutes to obtain a purified quartz extract and to remove the outer layer exposed to alpha irradiation. After HF treatment, the quartz grains were cleansed in HCl for 1.5 hours, rinsed, and then sieved over a 180 µm sieve to remove grains that were severely affected by the HF treatment.

For the measurement of the dose rate, the material removed from the outer 3 cm of the sampling tubes was combined with sample-adjacent material collected in plastic bags. The materials were dried at 105 °C for more than 12 hours to determine gravimetric moisture content. The organic matter content was measured by calculating the loss on ignition at 500 °C. The remaining materials were ground using a ball mill to a particle size smaller than 300 µm. The ground material was mixed with wax in a 70 : 30 sediment : wax ratio and was moulded into 2 cm thick pucks. The puck was analysed by the gamma spectrometer to measure the activity of $^{40}$K and several isotopes in the $^{238}$U and $^{232}$Th decay chains. The dose rate was calculated from the radionuclide concentrations following Guérin et al. (2011). We assumed an internal K-content of $10 \pm 2$ % for the K-feldspar grains to calculate the internal dose rate (Smedley et al., 2012). Since etching was not performed for the K-feldspar

grains, the external alpha contribution was taken into account with the assumption of $0.050 \pm 0.025$ Gy/ka. The cosmic radiation contribution to the dose rate was determined following Prescott and Hutton (1994), assuming gradual burial of the samples between deposition and present. Beta dose attenuation correction for the used grain size was performed according to Mejdahl (1979). Dose rate attenuation due to water and organic contents was taken into account following Aitken (1998). The full data essential to calculate the dose rate are provided in Supplementary Material A.

### 3.3 Luminescence measurements

Automated luminescence readers (Risø TL/OSL DA-15) equipped with $^{90}Sr/^{90}Y$ beta source, Blue-LED diodes, and IR-laser (Bøtter-Jensen et al., 2000; Bøtter-Jensen et al., 2003) were used for all measurements. All measurements (including gamma spectrometry) were performed at the laboratory of the NCL, at Wageningen University.

### 3.3.1 Quartz OSL

The quartz was positioned on stainless steel discs, placed within a circle with a 2 mm diameter ($\sim$ 50 grains / disc) on the centre of the discs using silicone oil ("Silkospray"). The measurement was conducted by applying the standard SAR protocol (Murray and Wintle, 2003) (see Supplementary Material B) with a preheat temperature of 200 ℃, and the stimulation temperature was 125 ℃. The OSL signal was measured for 20 seconds, the first 0–0.5 s interval was used for analysis, and the subsequent interval of 0.5–1.76 s was used for early background subtraction (Cunningham and Wallinga, 2010).  To determine the temperatures to be used for preheat and post-measurement bleaching, a thermal transfer (TT) test was performed (Truelsen and Wallinga, 2003). In the TT test, we first removed the luminescence by bleaching with the blue LED stimulation for 600 s in total. Subsequently, we gave the preheat for 10 s from 160 ℃ to 300 ℃ in 20 ℃ intervals. After each preheat, OSL signals were measured. We summed up the luminescence signals from each measurement to obtain a cumulative dose transferred from the preheat.

At the end of each SAR cycle, aliquots were bleached with Blue-LED for 40 seconds at an elevated temperature of 210 ℃ to completely reset the OSL signals. The acceptance of the results was based on the following criteria: 1) the recycling ratio should be smaller than 10 % of the unity; 2) the recuperation value smaller than 10 % of the largest regenerated signal; 3) the test dose error should be smaller than 10 %; 4) the IR signal should be less than 20 % of the total OSL signal, or the decrease of OSL signal after the exposure to IR should be less than 10 %. Errors were incorporated into the acceptance criteria.

A dose recovery test was performed to examine whether the samples could reproduce a given dose through the SAR procedure. For the dose recovery test, we first removed the luminescence with exposure to blue LED lights for 600 s in total and then gave each aliquot a dose of approximately 3.5 Gy.

### 3.3.2 Feldspar pIRIR

The prepared feldspar grains were placed on aluminium single-grain discs with 300 μm diameter holes arrayed in a 10 × 10 grid, each holding up to 100 grains. The grains were stimulated for 2 seconds with a 150 mW 830 nm IR laser. To select the

emissions from K-rich feldspars around 410 nm, a LOT/Oriel D410/30 interference filter was utilized. A SAR measurement protocol was adopted for De estimation, largely based on the pIRIR protocol proposed by Thomsen et al. (2008), which has been modified for single-grain measurements, in accordance with the observations made by Reimann et al. (2012) and Brill et al. (2018).

Prior to the measurement of the De, tests on remnant doses, dose recovery ratios, and fading rates were performed on feldspar grains that were bleached in a Hönle SOL2 solar simulator for 48 hours. Two samples were selected for the tests, one from the Cg horizon (NCL-1117123), and one from the 2Aap horizon (NCL-1117129). The tests incorporated three different pIRIR stimulation temperatures (150 ℃, 175 ℃, 225 ℃), to inform an appropriate stimulation temperature for this work. The tests on remnant doses and dose recovery ratios were performed on a single-grain level, and the fading rates were measured based on multi-grain aliquots (Auclair et al., 2003). Ultimately a preheat of 200 ℃ for 120 s and a pIRIR stimulation temperature of 175 ℃ were selected (see Results, section 4.2.).

We expected two types of scenarios for soil mixing of the plaggen deposits: (1) slow and less intense mixing by natural bioturbation decreasing with depth (Román-Sánchez et al., 2019; Wallinga et al., 2019) (Fig. 2.c), and (2) rapid and intense mixing of the upper 20-25 cm through ploughing during agricultural activities (van der Meij et al., 2019) (Fig. 2.d). These contrasting effects of different dynamics are also demonstrated in the results of von Suchodletz et al. (2023). However, since the plaggic anthrosols also incorporate a rapid deposition rate of materials, the samples collected from the plaggen layers were expected to contain a significant amount of poorly bleached grains due to slow bleaching of pIRIR signals and potentially short light exposure during soil mixing processes. Therefore, a method was devised to identify well-bleached grains. The adopted approach utilizes the different bleaching rates between the IRSL and pIRIR signals. It has been reported in previous studies that IRSL signals bleach much faster than pIRIR signals (Buylaert et al., 2012; Kars et al., 2014b). Therefore, if a grain is sufficiently light-exposed to fully reset the pIRIR signals, the IRSL signals will also be fully reset. This implies that well-bleached grains should provide matching $D_e$ values for the pIRIR and IRSL signals. In contrast, if a grain is only briefly exposed to light, we expect the IRSL signal to be better bleached than the pIRIR signal. This would result in a greater pIRIR De value compared to that of IRSL. Comparison of feldspar $D_e$ values at different temperatures has previously been proposed as a sediment tracer (e.g., Reimann et al., 2015; Chamberlain et al., 2017) and for identifying well-bleached samples with single-aliquot approaches (Buylaert et al., 2013) but it has not yet been used to identify well-bleached grains on a single-grain level for dating purposes.

Based on the above, a robust comparison between the $D_e$ values of IRSL signals, and the $D_e$ of pIRIR signals was made by the ratio of the two ($D_{e\ IRSL}$ / $D_{e\ pIRIR}$ ). Considering that IRSL signals are more prone to fading, we used a ratio of 0.9 as the threshold rather than unity. If the ratio is greater than 0.9 within a 2-σ error, we accepted the grain to be well-bleached. If it is smaller than 0.9 within the 2-σ error, the grain was classified as poorly bleached, and therefore rejected from age modelling of the feldspar single-grain datasets. While we acknowledge that the value of 0.9 is arbitrary, it was also adopted by Buylaert et al. (2013) for comparison of IRSL and pIRIR$_{290}$ signals. In this study, we use the term "filtering" to refer to the differentiation of well- versus poorly bleached grains using the $D_{e\ IRSL}$ / $D_{e\ pIRIR}$ ratio described above.

## 3.4 Age models

The soil at Braakmankamp is likely to have been exposed to a prolonged soil reworking process by agricultural activities and bioturbation (Fig. 2). Therefore, we applied two different age models to obtain ages that capture different phases of the soil reworking process. To determine the latest surfacing event of grains by soil reworking, this research applied the bootstrapped minimum age model (BsMAM) suggested by Cunningham and Wallinga (2012), to both quartz and feldspar $D_e$ datasets. For single-grain feldspar samples, to extract the grains with depositional information from the soil samples, we used the maximum age model (MaxAM) suggested by Olley et al. (2006). Combining the ages obtained by the BsMAM and MaxAM can provide a full narrative on the formation history of the soils with the soil reworking process taken into account. Evaluating between the quartz and feldspar datasets was done by comparing ages, which was achieved by dividing the individual $D_e$ of each aliquot / grain by the sample-average mineral-specific dose rate.

The chosen overdispersion input, or $\sigma_b$ value, can have a significant effect on the outcome of the BsMAM (Chamberlain et al., 2018; Cunningham and Wallinga, 2012). The $\sigma_b$ value indicates the characteristic overdispersion within a $D_e$ dataset for well-bleached sediments within a certain environment and for a given number of grains per disk (Cunningham and Wallinga, 2012). To determine the value of $\sigma_b$ to be applied to the BsMAM, this research adopted the method proposed by Chamberlain et al. (2018). This method obtains the characteristic overdispersion of well-bleached and un-mixed samples within the dataset by applying the BsMAM with $\sigma_b = 0 \pm 0$ to the relative overdispersion values obtained for the samples using the central age model (CAM) (Galbraith et al., 1999), and assumes that at least some samples within the dataset are well bleached. We determined the value by rounding the outcome of BsMAM applied to the relative overdispersion to the second decimal place.

## 4 Results

### 4.1 Quartz OSL tests

The average result of the thermal transfer test on the quartz samples demonstrates that the average values of thermally transferred doses agree with 0 Gy within 1-$\sigma$ standard error up to 300 °C (Fig. 4.a). However, the individual aliquots demonstrate greater scatter in the thermally transferred dose, and the majority of the aliquots are affected by preheat temperatures greater than 240 °C (Fig. 4.a). Therefore, we adapted the preheat temperature of 200 °C, and the temperature of 210 °C for the bleaching at the end of SAR sequence.

The dose recovery test of the SAR sequence for quartz demonstrated that the quartz aliquots were able to recover $D_e$ close to the given dose (CAM dose recovery ratio: $0.98 \pm 0.01$; given dose 3.56 Gy). The $\sigma_b$ input for quartz multigrain aliquots was determined to be $0.15 \pm 0.04$ following the method introduced by Chamberlain et al. (2018), applying BsMAM with the $\sigma_b$ input of $0 \pm 0$ to the relative overdispersion of the quartz OSL dataset.

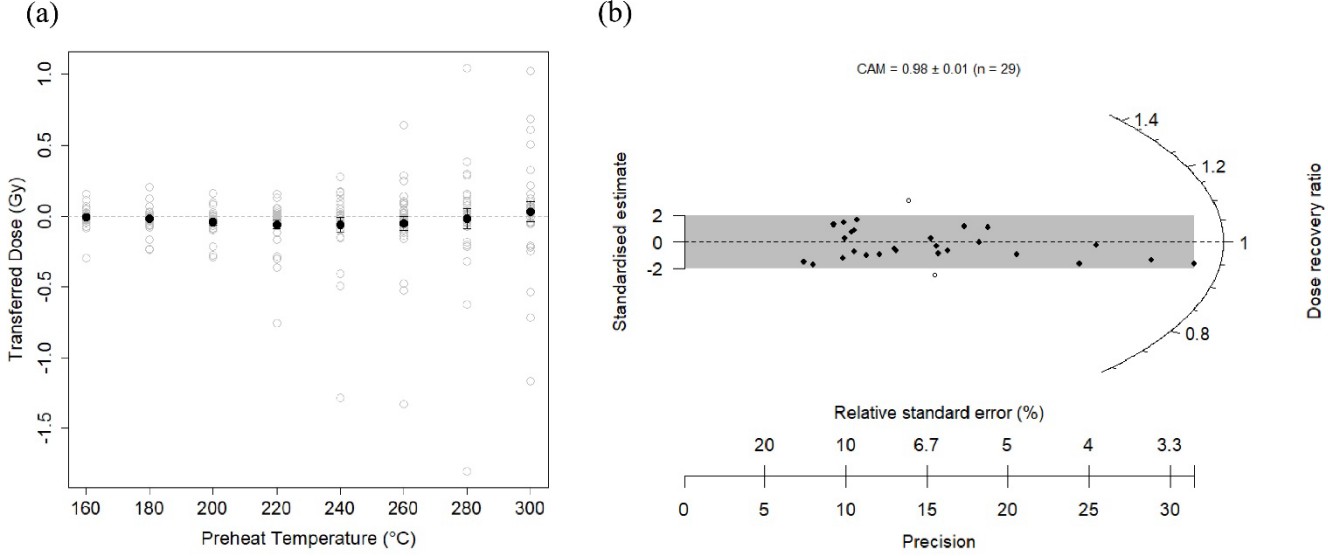

(a)

(b)

**Figure 4: The results of (a) thermal transfer test; (b) dose recovery test on quartz samples from the Braakmankamp site.**

### 4.2 Feldspar pIRIR tests

We found that remnant doses after 48 h SOL2 bleaching increased with preheat temperature and temperature for pIRIR stimulation (Fig. 5.a). As the $IRSL_{50}$ remnant doses increase as well, but less than those obtained through pIRIR, we attribute
the increase to a combination of thermal transfer (caused by preheating) and lower bleachability of high-temperature pIRIR signals (Kars et al., 2014b; Reimann et al., 2012).

The dose recovery tests on the pIRIR signals yielded ratios within 5 % from unity for all tested pIRIR temperatures (Fig. 5.b). In contrast, the dose recovery ratio of the $IRSL_{50}$ signals showed satisfactory results for the $pIRIR_{150}$ and $pIRIR_{175}$ sequences but underestimated severely for the $pIRIR_{225}$ sequence (Fig. 5.b). This result corroborates the finding by Kars et al. (2014a)
325 and is possibly attributed to trapping sensitivity changes occurring at high-temperature preheat (Wallinga et al., 2000).

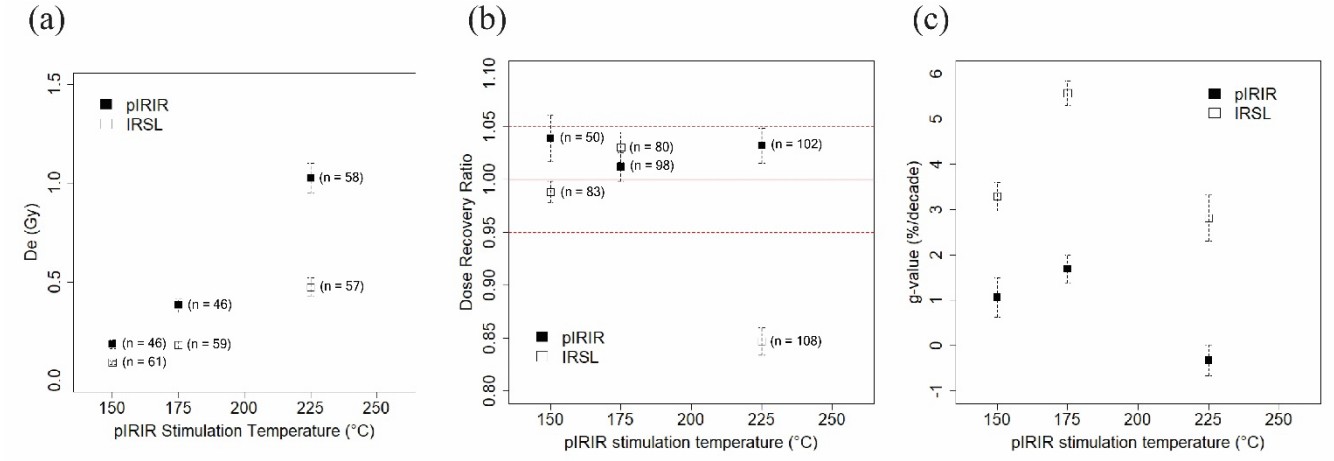

**Figure 5 The results of (a) remnant dose after bleaching in a solar simulator; (b) dose recovery test; (c) fading rate with different preheat / stimulation temperatures. The tests were performed on samples NCL-1117023 and NCL-1117029 after bleaching in the SOL2 solar simulator for 48 hours. The number of grains used for the analyses is indicated next to each data point (a, b). Six aliquots were used for fading analyses for each pIRIR stimulation temperature (c).**

For fading rates, the results show a lower correlation with the temperature used for pIRIR stimulation. It was demonstrated that the lowest g-value was obtained when the highest temperature (225 °C) was applied (Fig. 5.c). The g-value of the $IRSL_{50}$ signals were all substantially higher than the pIRIR signals, as expected. The g-values for the $pIRIR_{150}$ and $pIRIR_{175}$ agreed within the 1-σ error range. Based on these results, we decided to apply the $pIRIR_{175}$ protocol for age estimation. The main reason was that the dose-recovery results were most stable in both $IRSL_{50}$ and pIRIR signals when the $pIRIR_{175}$ protocol was applied, and this is important since we are using $IRSL_{50}$ and pIRIR signals for comparison. The $σ_b$ input for the single-grain feldspar $pIRIR_{175}$ was determined to be $0.35 ± 0.03$ for the complete dataset and $0.20 ± 0.04$ for the 'filtered' dataset.

### 4.3 Identifying well-bleached feldspar grains

When the comparison between $D_{e\ IRSL}$ and $D_{e\ pIRIR}$, or 'filtering', was performed, we observed that $D_{e\ pIRIR}$ values tended to be larger than $D_{e\ IRSL}$ values as expected (example provided in Fig. 6.a). There are a few grains that have larger $D_{e\ IRSL}$ values than $D_{e\ pIRIR}$ values, but for the majority of the grains, the $D_{e\ IRSL}$ / $D_{e\ pIRIR}$ ratio is smaller than or agrees with unity within a 2-σ error margin. Before examining the results of the comparison between $D_{e\ IRSL}$ and $D_{e\ pIRIR}$ values, we analyse the acceptance ratio as a function of depth (a proxy of age) to ensure that fading is not significantly affecting the results of the filtering. Since older samples are generally more affected by anomalous fading than younger ones, it is expected that fading would result in rejecting greater portions of grains for the deeper samples. However, there is a weak opposite trend within the depth and the acceptance ratio relationship (Fig. 6.b). When linear regression is applied, a positive relationship is observed. However, the statistical significance of the regression is relatively weak (p-value = 0.03).

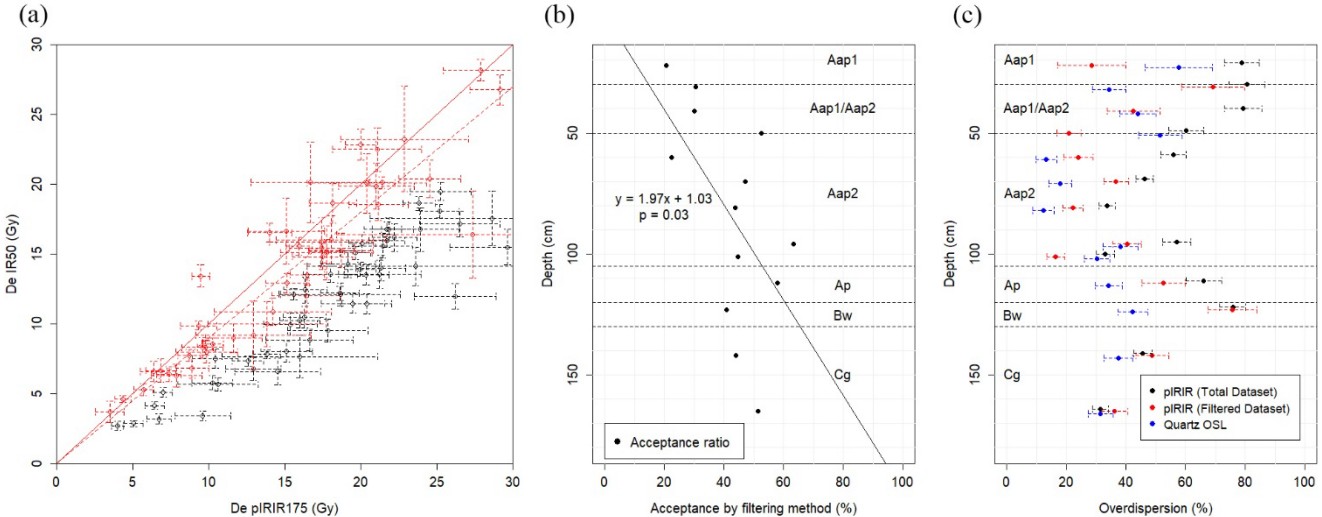

**Figure 6: (a) Example of grains characterized as well-bleached (red) versus poorly bleached (black) through comparison between $D_{e\,IRSL}$ and $D_{e\,pIRIR}$ values for sample NCL-1117123. The solid diagonal line marks the 1:1 ratio between equivalent doses obtained from $IRSL_{50}$ and $pIRIR_{175}$ signals. The dashed diagonal line marks the 1:0.9 ratio adopted as a threshold to identify well-bleached grains. (b) Acceptance ratio indicating the percentage of well-bleached grains with depth. The line demonstrates the trend between the relationship between depth and acceptance ratio obtained by linear regression. (c) Comparison of relative overdispersion obtained from the total dataset of feldspar $pIRIR_{175}$ (black), filtered dataset of feldspar $pIRIR_{175}$ (red), and quartz OSL (blue). Note that the points are slightly displaced vertically for visibility.**

Filtering using the $D_{e\,IRSL}$ / $D_{e\,pIRIR}$ ratio threshold impacted the $D_e$ distribution of most samples, with the strongest effect on the samples from the plaggen layer (Aap1, Aap1/Aap2, and Aap2 horizons). The overdispersion is reduced through the filtering approach for all samples from the Ap and Aap horizons, while negligible change is observed for samples from the Cg and Bw horizons (Fig. 6.c). A closer inspection of the $D_e$ distribution shows that the reduced overdispersion for samples from the Ap and Aap horizons results from the preferential rejection of older grains. For the Bw and Cg horizon samples, rejected grains show no bias, which explains why the overdispersion remains unchanged (see Supplementary Material C for a full observation of the results).

When comparing the ages obtained from BsMAM before and after applying the filter, the effect of the filtering can be observed, with only 2 out of 13 samples showing agreeing ages with a 1-σ error range (Fig. 7.a). While the agreement does not occur on the deeper samples collected below the Ap horizon (NCL-1117122 to NCL-1117124), the difference becomes more significant toward the present-day surface within the Aap1 and Aap2 horizons (that is, within the upper 1 m of the soil profile, Fig. 7.b).

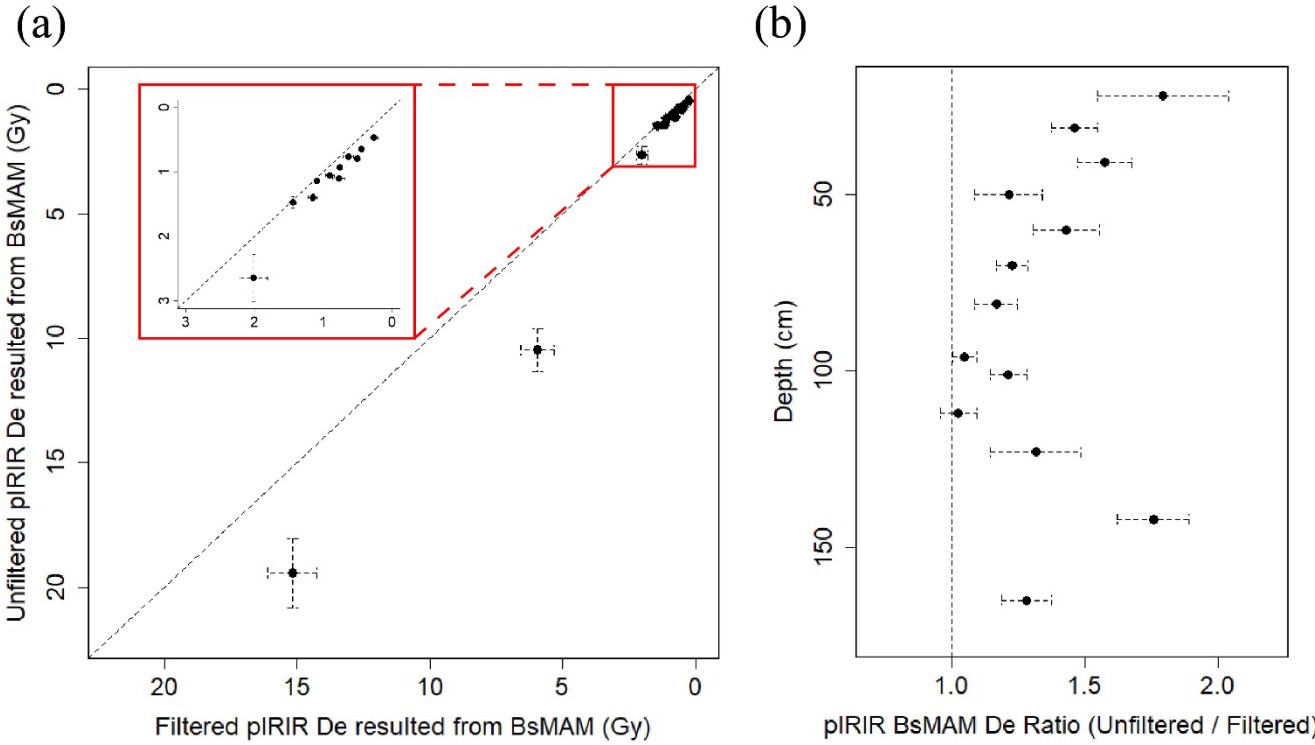

**Figure 7: (a)** Comparison of $D_e$ obtained by applying BsMAM to the filtered pIRIR dataset and the unfiltered pIRIR dataset. The dotted diagonal line represents unity. **(b)** The comparison between depth and ratio obtained by dividing the $D_e$ of the total pIRIR dataset by the $D_e$ of the filtered pIRIR dataset. The dotted vertical line represents unity. The majority of the samples (11 out of 13) demonstrate smaller $D_e$ when the filter is applied.

### 4.4 Results of the age model

For age modelling, we applied the BsMAM to the small-aliquot quartz datasets with a $\sigma_b$ input of $0.15 \pm 0.04$ and to the filtered single-grain feldspar pIRIR $D_e$ datasets with a $\sigma_b$ of $0.20 \pm 0.04$. The resulting palaeodoses were divided by the mineral-specific dose rate for each sample to obtain age estimates. Ages are reported in years (a) relative to 2017 (Table 2). Both sets of ages are internally consistent showing older ages for deeper sediments, with the exception of the feldspar pIRIR result at 70 cm depth. Quartz and feldspar agree for nearly all samples from the Aap horizons. For the horizons below, age results for feldspar tend to be lower compared to those for quartz.

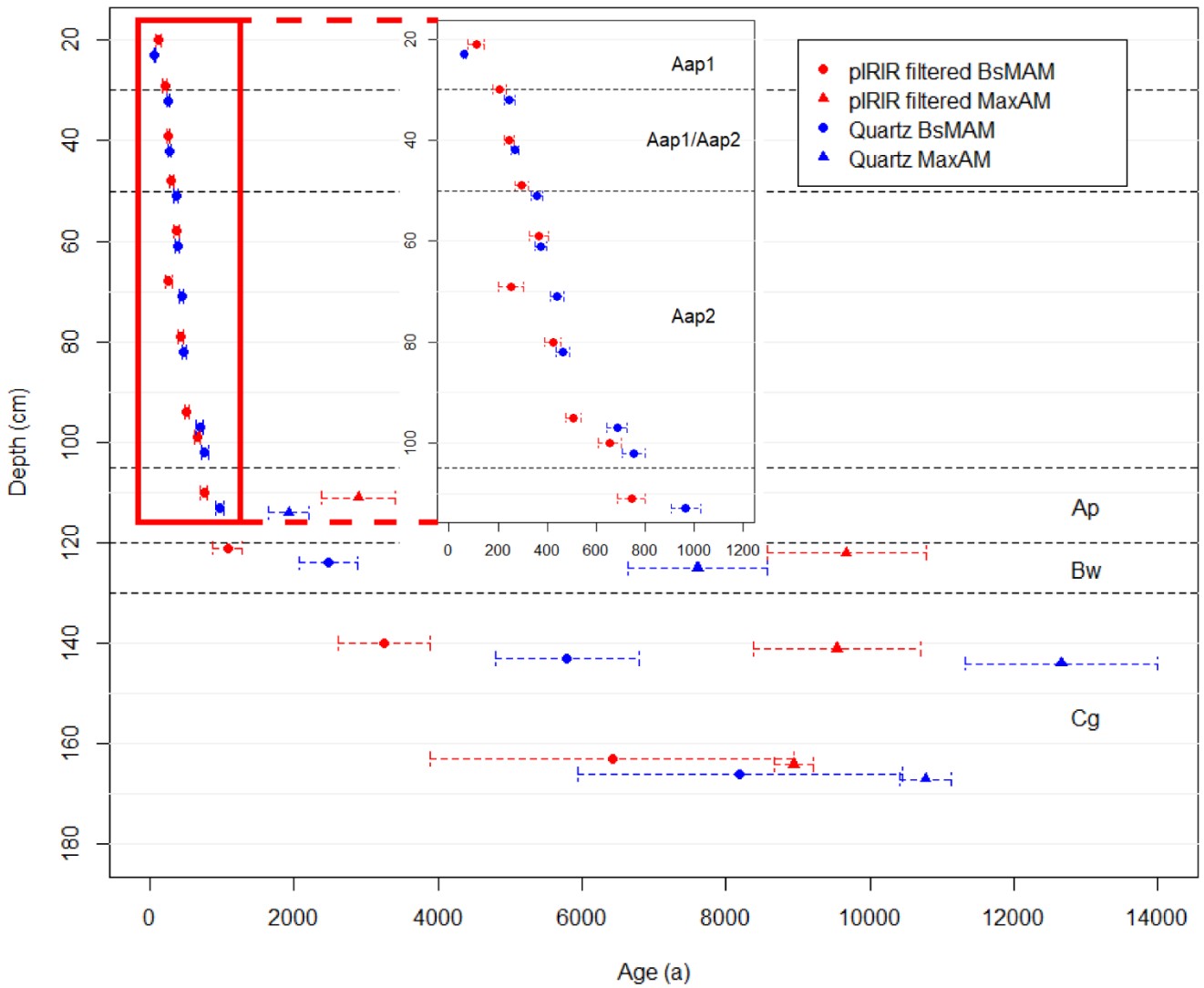

**Figure 8: Comparison between ages obtained by applying BsMAM to multigrain quartz (blue) and filtered single-grain feldspar pIRIR$_{175}$ (red) D$_e$ datasets. Note that the points are slightly displaced from their original position (quartz – bottom; feldspar – top) for the sake of visibility.**

**Table 2: Results of applying BsMAM and MaxAM to ages obtained from quartz signals and filtered ages from feldspar pIRIR$_{175}$ signals. The reference year of the date is 2017 when sampling and preparation of samples were conducted. Note that the ages are rounded to decades.**

| Sample | Depth (cm) | Quartz BsMAM (a) | Quartz MaxAM (a) | Feldspar (filtered) BsMAM (a) | Feldspar (filtered) MaxAM (a) |
|---|---|---|---|---|---|
| NCL-1117134 | 22 | 60 ± 5 (1960 ± 5 CE) | - | 110 ± 30 (1910 ± 30 CE) | - |

| NCL-1117133 | 31 | 240 ± 20 (1770 ± 20 CE) | - | 200 ± 30 (1810 ± 30 CE) | - |
|---|---|---|---|---|---|
| NCL-1117132 | 41 | 260 ± 20 (1750 ± 20 CE) | - | 240 ± 20 (1770 ± 20 CE) | - |
| NCL-1117131 | 50 | 350 ± 30 (1660 ± 30 CE) | - | 290 ± 30 (1720 ± 30 CE) | - |
| NCL-1117130 | 60 | 370 ± 20 (1640 ± 20 CE) | - | 360 ± 40 (1650 ± 40 CE) | - |
| NCL-1117129 | 70 | 440 ± 30 (1580 ± 30 CE) | - | 250 ± 50 (1760 ± 50 CE) | - |
| NCL-1117128 | 81 | 460 ± 30 (1550 ± 30 CE) | - | 420 ± 30 (1590 ± 30 CE) | - |
| NCL-1117127 | 96 | 680 ± 40 (1330 ± 40 CE) | - | 500 ± 30 (1510 ± 30 CE) | - |
| NCL-1117126 | 101 | 750 ± 50 (1260 ± 50 CE) | - | 650 ± 50 (1360 ± 50 CE) | - |
| NCL-1117125 | 112 | 960 ± 60 (1050 ± 60 CE) | 1920 ± 290 (100 ± 290 CE) | 740 ± 60 (1270 ± 60 CE) | 2890 ± 520 (870 ± 520 BCE) |
| NCL-1117124 | 123 | 2460 ± 410 (450 ± 410 BCE) | 7600 ± 970 (5580 ± 970 BCE) | 1070 ± 210 (940 ± 210 CE) | 9660 ± 1110 (7640 ± 1110 BCE) |
| NCL-1117123 | 142 | 5790 ± 1000 (3780 ± 1000 BCE) | 12650±1330 (10630 ± 1330 BCE) | 3240 ± 640 (1230 ± 640 BCE) | 9540 ± 1160 (7520 ± 1160 BCE) |
| NCL-1117122 | 165 | 8190 ± 2260 (6180 ± 2260 BCE) | 10770 ± 350 (8750 ± 350 BCE) | 6410 ± 2530 (4400 ± 2530 BCE) | 8940 ± 270 (6920 ± 270 BCE) |

## 5 Discussion

### 5.1 Identifying well-bleached grains for feldspar single-grain pIRIR dating

In this research, we used the ratio of $D_{e\ IRSL}$ and $D_{e\ pIRIR}$ values as a means to identify well-bleached grains within samples. From the results, we observed that there is a distinction between the population of grains that have been classified as well-bleached and poorly bleached. While the well-bleached grains are concentrated on the youngest population of the samples, the older grains consist of grains that have been classified to be poorly bleached (NCL-1117126 to 134, see Supplementary Material C). The division of ages between the two classes of grains supports that the ratio of $D_{e\ IRSL}$ and $D_{e\ pIRIR}$ values does

reflect the relative exposure time of grains to sunlight prior to being removed from exposure to sunlight. We checked the potential impact of anomalous fading by performing a regression analysis between depth and the acceptance ratio (Fig. 6.b).

The hypothesis is that if the results are affected by fading, older samples would have grains that would be falsely classified as poorly bleached since grains with higher doses are more susceptible to fading, as can be seen in Buylaert et al. (2012). However, the result of the analysis shows an opposite trend, which indicates that anomalous fading has insignificant effects on the results based on the ratio of $D_{e\ IRSL}$ and $D_{e\ pIRIR}$ in this research. This result is likely to have been caused due to the relatively young age of the samples, though to what degree fading affects the results should be further researched.

The results demonstrate that identifying and selecting well-bleached grains can have a crucial effect on the output of age models. Comparing the ages obtained by BsMAM on both filtered / non-filtered datasets shows that there is an average of 30 % overestimation on the ages from the non-filtered dataset when they are compared with the filtered ages, which we take to be most representative (Fig. 7.b). The main reason for the discrepancy in the ages is the difference in the selected $\sigma_b$ values that were applied to the BsMAM. In this research, we applied the method suggested by Chamberlain et al. (2018) to assume

that the lowest overdispersion of the dataset represents the overdispersion of a well-bleached sample. However, for the pIRIR signals in our sample site, none of the samples are completely well-bleached. Rather all samples include at least some poorly bleached grains. This is also supported by the high proportion of poorly bleached grains within the samples collected from the plaggen layer, where some have less than a 30 % acceptance ratio when the threshold for the $D_{e\ IRSL}$ / $D_{e\ pIRIR}$ ratio is applied (Fig. 6.b). The high percentage of poorly bleached grains yields high overdispersion values (Fig. 6.c), and thereby

overestimates the $\sigma_b$ value that should be applied. Better agreement is seen between the OSL and the un-filtered pIRIR ages when the $\sigma_b$ value of 0.20 ± 0.04 obtained from the filtered dataset of grains that have been classified as well-bleached grains, is used as input to the BsMAM for pIRIR age modelling (Table 3). This indicates that using the ratio of $D_{e\ IRSL}$ and $D_{e\ pIRIR}$ values can be a useful way to determine the $\sigma_b$ value for feldspar samples.

**Table 3: Comparison of BsMAM results between the dataset of total grains using two different $\sigma_b$ values and the dataset of grains**
**classified to be well-bleached.**

| Sample | Depth (cm) | BsMAM result (Gy) pIRIR$_{175}$ total dataset $\sigma_b = 0.35 \pm 0.03$ | BsMAM result (Gy) pIRIR$_{175}$ total dataset $\sigma_b = 0.2 \pm 0.04$ | BsMAM result (Gy) pIRIR$_{175}$ filtered dataset $\sigma_b = 0.2 \pm 0.04$ |
|---|---|---|---|---|
| NCL-1117134 | 22 | 0.28 ± 0.05 | 0.24 ± 0.04 | 0.23 ± 0.07 |
| NCL-1117133 | 31 | 0.55 ± 0.09 | 0.36 ± 0.13 | 0.41 ± 0.05 |
| NCL-1117132 | 41 | 0.70 ± 0.06 | 0.59 ± 0.08 | 0.50 ± 0.03 |
| NCL-1117131 | 50 | 0.75 ± 0.03 | 0.71 ± 0.07 | 0.61 ± 0.05 |
| NCL-1117130 | 60 | 1.09 ± 0.04 | 1.03 ± 0.08 | 0.77 ± 0.07 |
| NCL-1117129 | 70 | 0.88 ± 0.10 | 0.65 ± 0.09 | 0.54 ± 0.10 |
| NCL-1117128 | 81 | 1.09 ± 0.04 | 0.97 ± 0.08 | 0.89 ± 0.05 |

| NCL-1117127 | 96 | 1.15 ± 0.04 | 1.09 ± 0.07 | 1.07 ± 0.04 |
| --- | --- | --- | --- | --- |
| NCL-1117126 | 101 | 1.51 ± 0.06 | 1.35 ± 0.09 | 1.31 ± 0.06 |
| NCL-1117125 | 112 | 1.49 ± 0.13 | 1.30 ± 0.29 | 1.46 ± 0.08 |
| NCL-1117124 | 123 | 3.38 ± 0.55 | 2.39 ± 0.32 | 2.16 ± 0.40 |
| NCL-1117123 | 142 | 12.08 ± 1.73 | 8.01 ± 1.44 | 6.78 ± 1.30 |
| NCL-1117122 | 165 | 19.40 ± 1.41 | 16.51 ± 3.37 | 13.82 ± 5.42 |

Notably, the overdispersion of 'well-bleached' feldspars is similar to that of quartz, resulting in similar $\sigma_b$ values for both minerals (Fig. 6.c). Smedley et al. (2019) reported a constant offset of ~ 10 % between the overdispersion between single-grain quartz and single-grain feldspar. The difference is greater between the small-aliquot quartz OSL and the non-filtered single-grain feldspar pIRIR dataset for the samples from Braakmankamp, which is ~ 20 %. Considering that aliquot-based measurements will have smaller overdispersion compared to single-grain measurements (Thomsen et al., 2012), this is within expectations. However, the overdispersion is significantly reduced by selecting well-bleached grains using the $D_{e\,IRSL}$ / $D_{e\,pIRIR}$ ratio. This indicates that feldspar single-grain measurements may have incorporated a significant amount of poorly bleached grains even for samples with well-bleached quartz OSL characteristics.

Another factor that should be considered with the overdispersions of OSL and pIRIR datasets is the influence of external microdosimetry. In the dose recovery experiments conducted for both minerals, it has been observed that the overdispersion of small-aliquot quartz OSL is typically 5–10 % (e.g. Thomsen et al., 2005), and single-grain feldspar pIRIR is 15–20 % (e.g. Brill et al., 2018; Reimann et al., 2012). Based on the dose-recovery experiments, the difference of ~ 10 % overdispersion is caused by intrinsic characteristics of the minerals, and cannot be explained by the inclusion of poorly bleached grains. However, the Braakmankamp samples with well-bleached quartz OSL characteristics do not demonstrate such a discrepancy. This may be an effect of external microdosimetry, while small-aliquot quartz is more influenced by external microdosimetrical variations than single-grain feldspar, which has significant influence by the internal K-content (Smedley et al., 2019). Thus, one possible scenario is that the external microdosimetry of the samples evens out the intrinsic ~ 10 % overdispersion difference by increasing the small-aliquot quartz OSL overdispersion to a more significant degree. However, the effects of external microdosimetry and internal K-content are not tested in our research and therefore are to be further researched.

## 5.2 Comparison of single-grain pIRIR$_{175}$, IRSL$_{50}$, and small-aliquot OSL ages

When comparing the ages obtained from the single-grain feldspar pIRIR dataset (selected grains using $D_{e\,IRSL}$ / $D_{e\,pIRIR}$ ratio filtering) with the quartz ages, we observe that five out of nine ages agree with each other within the 1-$\sigma$ error range. For two additional samples, the differences between ages are minor, that is, they agree within 2-$\sigma$. Most of the samples with agreeing ages are concentrated at the upper part of the plaggen layer, besides the top sample (NCL-1117134). From the samples from

the lower part of the plaggen layer (NCL-1117126 & 127), the pIRIR signals produced younger ages than quartz OSL. This is also the case for the samples collected from the Bw and Cg horizons below the plaggen layer.

The disagreement in ages in the samples collected from the deeper layers beneath the plaggen layer may be largely due to the averaging of multiple quartz grains within aliquots. When we examine the outcome of the BsMAM and the MaxAM of the samples NCL-1117124 and NCL-1117125, it is observed that the age range obtained from BsMAM and MaxAM being applied to quartz OSL is within the age range derived by applying BsMAM and MaxAM to pIRIR signals (Table 2, Fig. 8). These samples seem to have heterogeneous age distribution, in which the youngest grains may be detected by single-grain dating methods, but not by dating methods based on aliquots.

The result of the youngest sample (NCL-1117134) demonstrates a different pattern. Unlike the other results, in which the ages obtained by BsMAM applied to pIRIR signals are younger than the ages from quartz signals, the age from pIRIR signals is older than that from quartz signals. This might have been caused by the presence of grains with hard-to-bleach yet relatively minor remnant doses with pIRIR signals (Kars et al., 2014b), which can also be observed by the test performed on grains bleached in the solar simulator (Fig. 5.b). However, it is difficult to conclude that this is the principal reason for the difference in NCL-1117134 since each individual grain can behave differently, which can also be seen in the NCL-1117134 sample itself, where grains that are bleached to near-zero dose are also present (Supplementary Material C). Near-zero dose bleached grains are also of low precision, meaning they have only a minor influence on the BsMAM.

Based on our results, multi-grain quartz OSL performed on small-diameter aliquots that provide a sufficient proxy for genuine single-grain quartz OSL measurements is a suitable approach for dating the plaggen layer because it appears to provide ages that are stratigraphically consistent and reflects the formation history of this anthropogenic layer. Also, there are well-bleached samples within the plaggen layer providing nearly identical results from both the central age model (CAM) and BsMAM (NCL-1117128, NCL-1117129, NCL-1117130, see Table 4). Since other dating methods are problematic for plaggic anthrosols, quartz OSL ages of the well-bleached samples can function as an age constraint to vet the more experimental luminescence approaches. The ages obtained from pIRIR signals mostly align with the quartz OSL ages in this study, showing the possibility that single-grain feldspar pIRIR dating can be useful for obtaining reliable ages and may be an important tool for dating anthropogenic soils in settings where quartz does not possess suitable characteristics for OSL dating. In contrast, the $IRSL_{50}$ ages (Supplementary Material D) only agree with the quartz OSL ages for one of the nine plaggen samples, while the majority of $IRSL_{50}$ ages are underestimated relative to the quartz results. These results suggest that fading is significantly affecting the $IRSL_{50}$ signals, despite the samples being young. Therefore, despite lower bleachability, pIRIR signals may be preferable over IRSL signals for the dating of late Holocene sedimentary deposits. The full results of $IRSL_{50}$ signals can be observed in Supplementary Material D. Finally, although time-consuming, our approach highlights the value of employing different luminescence signals and minerals to enhance confidence in dating results for challenging materials such as soils.

**Table 4. CAM and BsMAM $D_e$ results obtained from quartz OSL signals. The radial plots of all samples are available in Supplementary Material E.**

| Sample | Depth (cm) | Quartz OSL CAM result (Gy) | Quartz OSL BsMAM result (Gy) |
|--------|-----------|----------------------------|------------------------------|

| | | | $\sigma_b = 0.2 \pm 0.06$ |
|---|---|---|---|
| NCL-1117134 | 22 | 0.12 ± 0.02 | 0.07 ± 0.01 |
| NCL-1117133 | 31 | 0.30 ± 0.02 | 0.26 ± 0.02 |
| NCL-1117132 | 41 | 0.37 ± 0.03 | 0.29 ± 0.01 |
| NCL-1117131 | 50 | 0.47 ± 0.05 | 0.41 ± 0.02 |
| NCL-1117130 | 60 | 0.44 ± 0.02 | 0.43 ± 0.02 |
| NCL-1117129 | 70 | 0.53 ± 0.02 | 0.52 ± 0.03 |
| NCL-1117128 | 81 | 0.54 ± 0.02 | 0.54 ± 0.03 |
| NCL-1117127 | 96 | 0.88 ± 0.07 | 0.75 ± 0.04 |
| NCL-1117126 | 101 | 0.87 ± 0.05 | 0.80 ± 0.04 |
| NCL-1117125 | 112 | 1.11 ± 0.07 | 1.00 ± 0.05 |
| NCL-1117124 | 123 | 3.71 ± 0.26 | 2.66 ± 0.43 |
| NCL-1117123 | 142 | 9.89 ± 0.65 | 6.64 ± 1.12 |
| NCL-1117122 | 165 | 12.15 ± 0.69 | 9.95 ± 2.72 |

## 5.3 Luminescence informed soil formation history

Beyond ages, luminescence dating can provide information on the formation history of soils (e.g. Wallinga et al., 2019). Especially the single-grain feldspar pIRIR signals can provide additional information on the formation process (e.g. Gray et al., 2020; Reimann et al., 2017; von Suchodoletz et al., 2023). The different age distribution patterns between the samples from plaggic anthrosols and natural layers demonstrate the effects of bioturbation and human activities on age distributions. Samples from the plaggen layers demonstrate a relatively larger proportion of poorly bleached grains, and a relatively small dispersion of age after being processed by the filtering method (Fig. 6.c). In contrast, a smaller proportion of grains are identified as poorly bleached for the natural layers even though the age distributions are overdispersed. We propose that this contrast can be explained by the difference between the effects of natural bioturbation and anthroturbation.

The difference in the overdispersion between the samples collected from plaggen layers and the samples collected from the underlying deposits can be explained by the change in the pedoturbation caused by the change in land use. The overdispersion of single-grain pIRIR $D_e$ values within the natural layers can be explained by the effects of natural bioturbation driven by a deeper soil food web, involving deep-rooted trees and an increased number of soil fauna that utilizes deep plant roots (Maeght et al., 2013) (Fig. 9.c). Such natural bioturbation is a constant but slow process, where grains that are brought to the surface will stay there for sufficient time to totally bleach luminescence signals (Bateman et al., 2007; Kristensen et al., 2015; Reimann et al., 2017; von Suchodletz et al., 2023). As a result, the material within this mixing zone will contain well-bleached grains of different luminescence ages. The age range will be wide due to the long period between deposition (oldest grains, late

Glacial) and end of bioturbation (youngest grains, plaggen agriculture introduction). In our data, the influence of bioturbation is visible within NCL-1117123 and not within NCL-1117122. This suggests that pre-agricultural bioturbation at Braakmankamp likely caused soil reworking to a depth of 30–50 cm below the pre-plaggen palaeosurface. Prior human land-use activities on this site (e.g. land clearance) before the start of agricultural activities are not evident in the luminescence dating results.

For the plaggic layer, natural bioturbation is replaced by intense anthroturbation through ploughing (Fig. 9.d). This would likely cause the soil food web to be shallower with the reduction of specific subterranean fauna due to the removal of deep roots (Maeght et al., 2013). A much greater part of the grain is now (repeatedly) brought to the surface, but light exposure duration is expected to be much shorter (Poręba et al., 2013; van der Meij et al., 2019; von Suchodletz et al., 2023). Although plaggic material may contain Pleistocene-aged material when first applied, light exposure of the vast majority of grains will result in bleaching of luminescence signals. Subsequent shallow reworking will mix grains that yield similar (young) luminescence ages in smaller age dispersion (Fig. 5.c). This shift toward different ecological (forest vs grassland) and land use (natural vs agricultural) regimes and thereby different soil mixing processes that we interpret in the pIRIR $D_e$ distributions and bleaching results is supported by the pollen analysis of the samples taken from the same site (Fig. 9.a). The pollen analysis demonstrates a significant decrease in pollen of trees and an increase in pollens of herbs and cereals, around 1000 years ago (Fig. 9.a) (Smeenge, 2020).

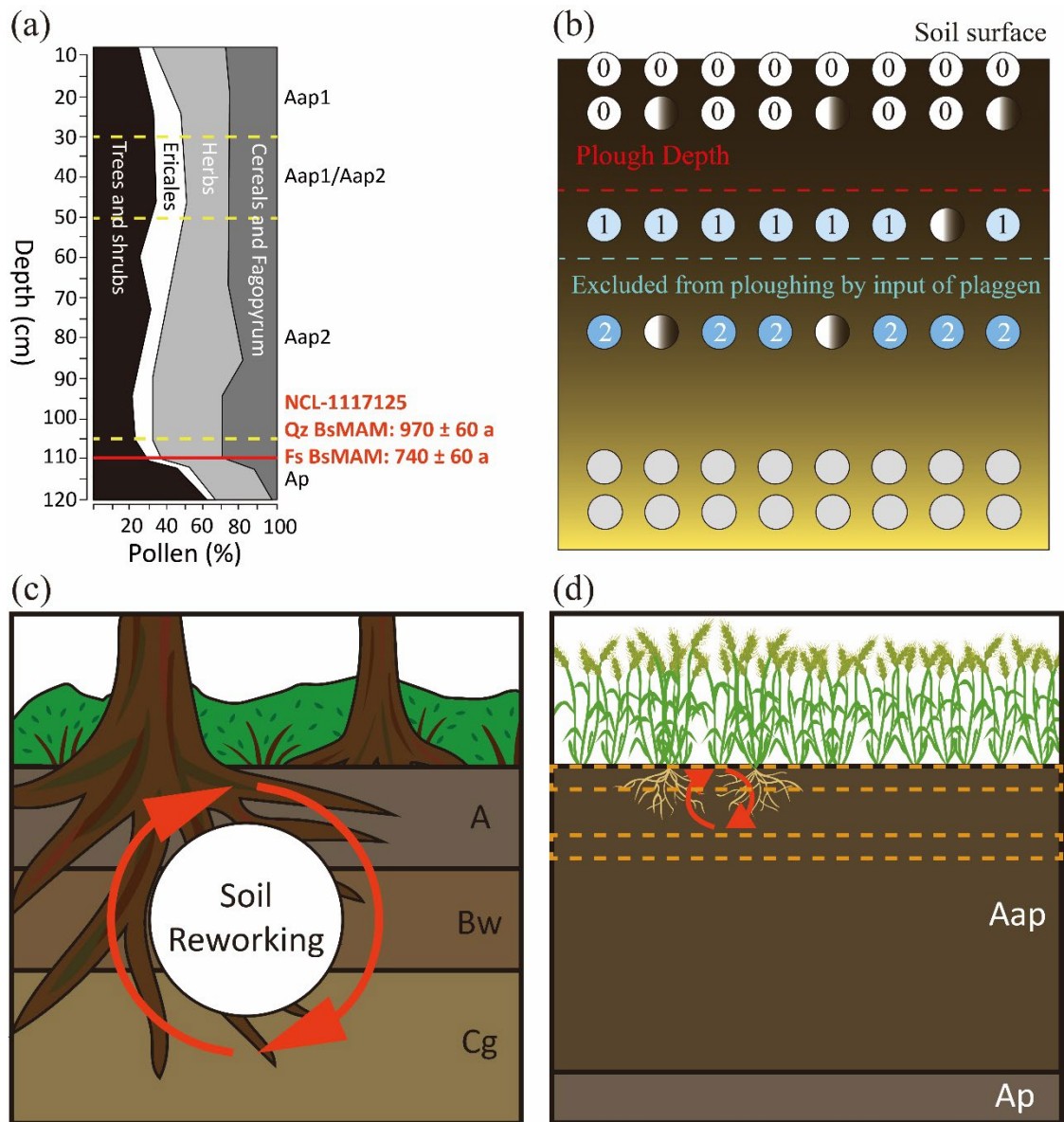

**Figure 9: (a) Simplified pollen analysis result of Braakmankamp from Smeenge (2020). A sharp decrease in trees and shrubs coincides with the increase in herbs, cereals, and Fagopyrum. (b) A conceptual diagram illustrating the expected luminescence signals of grains within the plaggen layer at Braakmankamp (see Fig. 2 for symbol reference). While the topsoil is continuously reworked by ploughing, a certain depth of the soil layer is excluded from the soil reworking by accretion of plaggen, creating a series of layers with grains of a certain age range. (c) Possible scenario of soil reworking before the introduction of plaggen agriculture at Braakmankamp. The deeper soil food chain, mainly deep-rooted trees, moves the grains from the surface to relatively deep layers. (d) Possible scenario of soil reworking after the introduction of plaggen agriculture at Braakmankamp. The decrease of the depth affected by the soil food chain, due to cereals and herbs with shallower roots and the decrease of soil fauna dependent on deep roots, cause decreased movement of grains. Also, the yearly addition of plaggen materials (indicated by the dashed box) results in previous deposits being excluded from the soil reworking process.**

The high proportion of poorly bleached grains in the plaggen layer can be interpreted as an outcome of the combination of intensive ploughing activities and increased depositional rate of materials after the introduction of plaggen agriculture. The intensive ploughing caused more grains to be exposed to sunlight, which can be seen in the lower overdispersion within the $D_e$ values of quartz signals (Fig. 6.c). However, the duration of the exposure for many of the grains would have been relatively short, compared to that before the introduction of plaggen agriculture, which resulted in a high proportion of feldspar grains with poorly bleached pIRIR signals (e.g., Fig. 2.d). There is an evident increase in depositional rate after the introduction of plaggen agriculture, which can also be observed in the depth-age plot (Fig. 8). When a simple calculation based on the depth and BsMAM age is made, the depositional rate after plaggen have been applied to the field plots is 1.4 mm/yr. The depositional rate of the layers under the plaggen layer cannot be derived the same way, since the BsMAM results are likely to be artefacts of bioturbation resetting (see Fig. 2.c), and would probably represent the time that the layers have been last exposed to pedoturbation. For these layers, the depositional rate remained small, resulting in prolonged exposure to soil reworking caused by bioturbation. However, after plaggen agriculture was introduced, the time that each grain was under the influence of soil reworking would have reduced, with approximately 14 cm of deposits being excluded from the cycle every century. This results in a series of layers consisting of grains with a similar age distribution with partial inclusion of poorly bleached grains, which is a distinctive pattern for plaggic anthrosols (Fig. 9.b). This result also aligns with the idea that the effects of agriculture are likely to superimpose the effects of natural bioturbation on soil reworking (von Suchodoletz et al., 2023).

### 5.4 Implications for the timing and formation of plaggic anthrosol on the Braakmankamp site

Ideally, multiple dating methods should be combined to obtain chronologies of landscape evolution, because this allows for cross-checking of assumptions and findings. Yet, the resulting datasets must be reconciled to obtain a single chronology for the site in question. Here, we use insights from both quartz and feldspar luminescence measurements, including dose distributions and comparisons of age datasets, to reconstruct the soil formation process at Braakmankamp. The comparison of multiple luminescence methods informs the selection of appropriate approaches for different deposit types and gives confidence in the assignment of time periods to the depositional events at Braakmankamp.

The first recorded event is the deposition of aeolian coversands, which occurred mainly around 9–10 ka ago according to the results of the MaxAM of feldspar (samples NCL-1117122 to 124). The aeolian sand functioned as the parent material for the brown forest soil that was formed on the site before the reclamation. We adopt the MaxAM to derive the depositional age to avoid underestimation of age due to bioturbation. It should be noted that bioturbation can work both ways, by introducing older grains from deeper layers (as in Fig. 2.c) and therefore causing an overestimation of ages when MaxAM is applied (Yates et al., 2024). However, in the specific context of this site, the last major depositional unit in the broader region (besides the deposition of driftsands, which occurred after the Middle Ages) is the Younger Coversand II observed in Lutterzand, Twente, which is a thick layer of coversand deposited in a relatively short period of time (Vandenberghe et al., 2013). The parent material of Braakmankamp is very much likely to correspond to this sedimentary unit. Therefore, we believe that the overestimation by the inclusion of older material is less likely since the materials would have originated from the same unit.

For the same reason, we prefer the single-grain pIRIR ages over single-aliquot quartz OSL ages. The ages obtained from the MaxAM results on feldspar are more consistent with each other than those of quartz. However, the pIRIR age may have been affected by fading, and the actual age of the deposition of the aeolian coversand is likely to be around 12 ka, which is obtained
from the quartz CAM of NCL-1117122. This corresponds with the ages of the Younger Coversand II, which was dated between $13.6 \pm 1.1$ ka to $12.2 \pm 0.9$ ka (Vandenberghe et al. 2013). The degree to which fading affects the discrepancy between the single-grain feldspar BsMAM and the multi-grain quartz BsMAM should be further researched.

The reclamation of the Braakmankamp site seems to have occurred around 900–1000 years ago (1020–1120 CE), which is around the 11th - 12th century CE. The quartz BsMAM result of the sample (NCL-1117125) from the Ap horizon is $970 \pm 60$
a. The feldspar BsMAM age for this sample is a bit younger, but the sample collected below the Ap horizon (NCL-1117124) gave the result of $1070 \pm 210$ a. Considering that the youngest population of the ages in this sample is likely to reflect the grains that have been reworked, the BsMAM result of the feldspar from the Bw horizon imposes that the soil reworking has affected the Bw horizon up to around 1000 a. The difference between the quartz and feldspar in these samples seems to have occurred because of the difference between aliquot-based dating and single-grain-based dating.

The introduction of plaggen agriculture to the site seems to have occurred around 700–800 years ago (1220–1320 CE) based on the quartz BsMAM results of NCL-1117126 (Table 2). A nearly identical age is obtained from the feldspar BsMAM results of NCL-1117125 (Table 2). Based on the results, after the introduction of plaggen agriculture, the intensity of the mixing has increased, allowing more grains to reach the surface. However, the depth of the mixing has decreased during this period, due to human land-use practices. The main source of mixing would have been limited to ploughing, which would not have affected
more than 20 cm of depth during the period (van der Meij et al. 2019). The rapid accumulation of plaggen materials would also preserve deeper layers from soil reworking with the accumulation rate of $\sim 1.14$ mm/yr, which is understood to be typical for the plaggic anthrosols in this area (Spek, 2004).

## 6 Conclusion

Establishing robust chronology and developing knowledge of the formation process of plaggic anthrosols are crucial to
understanding the landscape evolution in Western Europe and human-landscape dynamics. Our study approaches this through a combination of established and new luminescence methods for 13 luminescence samples collected from a plaggic anthrosol site in Braakmankamp, eastern Netherlands. This yields insights into both the formation history of an anthropogenic soil and luminescence approaches for quantifying soil formation. Our main observations are:

– The ratio of $D_{e\,IRSL}$ and $D_{e\,pIRIR}$ values for single-grains is a new tool to largely exclude poorly bleached grains from
luminescence age estimations and obtain robust overdispersion values for well-bleached grains, which are needed for minimum age modelling.

– Small-aliquot quartz OSL is viable for dating the Braakmankamp and likely other plaggen deposits. Single-grain feldspar pIRIR yields similar ages as single-aliquot quartz OSL when filtered by the $D_{e\,IRSL}$ / $D_{e\,pIRIR}$ ratio. The filtered pIRIR ages are

more valid than those obtained from $IRSL_{50}$. For underlying natural layers that experienced significant bioturbation, single-grain feldspar pIRIR is capable of providing the ages related to the soil-reworking to the full extent.

– Single-grain feldspar pIRIR data reflect bioturbation history, making it possible to reconstruct the transition between forested and agricultural landscape regimes in the soil column.

– At Braakmankamp, site reclamation involving land clearance occurred around 900–1000 years ago and plaggen agriculture began around 700–800 years ago.

In summary, this research demonstrates that combining single-grain feldspar pIRIR and small-aliquot quartz OSL measurements can be a useful tool in dating and reconstructing soil formation processes. The application of single-grain feldspar pIRIR measurements has added value in understanding the dynamics of different sources of pedoturbation. Overall, this research provides a methodological approach to luminescence dating and an example of its application on anthrosols, which will be useful for the reconstruction of anthropogenic landscapes elsewhere.

## Data availability

The data and metadata are available at https://doi.org/10.4121/4ad5c9b3-f0f1-4757-b845-681268a707e4.

## Author contributions

JC, RvB, ELC, TR, HS, AvO, and JW contributed to the conceptualization of different aspects of the research. TR, HS, and AvO conducted the fieldwork and sampling. Funding acquisition was undertaken by JW, TR, and RvB. HS carried out the main investigation on the research site, including site selection, soil profile, and pollen analysis. JC, and AvO performed luminescence measurements and analysed the data, supported by TR, ELC, and JW. JC prepared the manuscript with contributions from all co-authors.

## Competing interests

The authors declare that they have no conflict of interest.

## Acknowledgements

This research is part of the EARTHWORK project funded by the NWO 'Archeologie Telt' programme (AIB.19.013). We appreciate Mr. Ben H. Nieuwe Weme for the permission to conduct fieldwork in the study site. We also thank for Alice Versendaal and Erna Voskuilen for their assistance during the lab work.

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
