# Peer review of "Luminescence dating approaches to reconstruct the formation of plaggic anthrosols"

_EGUsphere, 2023_

## Referee Comment (RC3)

This paper is an extensive study using different luminescence signals to investigate and date plaggic anthrosols at a site in The Netherlands. The paper is data-rich, well-written and interesting to read. Samples are generally young (hundreds to thousand years) and soil mixing processes are active so the challenge is to interpret $D_e$ distributions in terms of bleaching and mixing, and to try to extract the correct $D_e$ from the $D_e$ distributions. I have one major comment on the methodology that the authors should address.

Major comments

1) From the application of a Minimum Age Model (here bootstrapped MAM) to both quartz OSL and feldspar pIRIR signals the reader would infer that both the feldspar pIRIR and the quartz OSL signals are partially reset or show (high) dose tails (possibly due to mixing). However, at least 3 samples (NCL-11171 28,29 &30) have an quartz OSL over-dispersion (OD) of ~15% (see Fig. 5) and this is identical to the input OD of a well-bleached sample for BsMAM modeling. So, one would consider these samples as well-bleached for quartz OSL. Do the BsMAM and the CAM (or weighted or unweighted means if the authors prefer) give the same answer as the BsMAM for these samples? If the BsMAM works for well-bleached material both models should return identical results and the authors should demonstrate this. The CAM results should be listed and compared with BsMAM.

Actually, the authors themselves allude on samples with well-bleached quartz OSL characteristics (see lines 394-395); please show the quartz OSL $D_e$ distributions for all samples in Supp Info.

Would it be possible to discuss the average $IR_{50}$ results in this paper too? If the $IR_{50}$ signal is sufficiently reset which is definitely possible for the three samples mentioned above, these samples are likely to give $IR_{50}$ ages smaller than quartz OSL (because of fading). The samples that are less-well bleached for both $IR_{50}$ and $pIRIR_{180}$ signals will tend to give ages equal to or larger than quartz OSL. I miss a discussion at the level of the average behaviour (CAM, weighted mean) in this manuscript.

This leads me to the proposed research question: 1) How can well-bleached grains be identified for feldspar single-grain pIRIR dating? In my view, to answer this question one needs some form of independent age control. I cannot find that in this paper, especially because the authors believe that the quartz OSL ages should also be inferred from a MAM approach. The best option in the case one does not have independent age control, would be to use a well-bleached, unmixed quartz age (from CAM) and compare with the MAM age of feldspar pIRIR (filtered or unfiltered).

2) I cannot seem to find the radionuclide concentrations, used water contents and the total dose rates in the paper. These data are crucial to calculate luminescence ages and should be tabulated.

Minor comments:

Suppl Mat A.1 (Table): suggest to change cutheat to preheat. Cutheat refers to immediate cooling after reaching temperature but test dose preheat here has duration of 10s.

Line 17: humans, remove second recently,

Line 36: has created

Line 39: factor in the creation of anthrosols?

Line 56 (caption): at Braakmankamp

Line 79: remove full stop after question mark

Line 102: northern

Line 145: place at a site

Line 159 (caption): in areas with coversand

Line 162: At all depths

Line 169: gleying

The K-feldspar grains were not etched. Did you take into account an external alpha contribution? If so, how large is it?

Line 246-247: These contrasting effects, von Suchodoletz

Line 252: Poolton et al. looked at elevated temperature IRSL but not post-IR elevated temperature IRSL? Please check, if not pIRIR, then remove ref.

Lines 285-286: not logical after previous sentence in which it is stated that TT is very small or negligible (at least I cannot see a trend). There is more scatter in the results but this is not necessarily due to thermal transfer? Can also be sensitivity changes not full accounted for by test dose? Please rephrase.

Line 324: majority of the samples

Line 343: Fig. 7a

Line 346: remove second full stop

---

## Author Comment (AC1)

| Comments | Response |
|---|---|
| **For sections** | |
| 1. The abstract is a little bit cumbersome, consider deleting the second and third sentences. | We shall revise the abstract and replace the mentioned sentences. |
| 2. The second section might be merged into the first section, placed just before presenting the scientific questions and research objectives. | We prefer to keep the introduction section relatively short, to help highlight the research questions and objectives. The subsequent 'dating plaggen soils' section is longer and more detailed because it provides background needed for readers of different expertise (archeological and luminescence communities) to grasp the study and its technical aspects. |
| **For figures** | |
| 1. Figure 2 should be labeled with latitude and longitude. | We shall add the grids and coordinates to the figure. |
| 2. The title of Figure 3 (b)'s z-coordinate should be changed to "Dose recovery ratio". | We shall change the label of the z-coordinate as recommended. |
| 3. In the caption of Figure 4, it should be specified that the average value is calculated from how many results. | We shall add the number of grains / aliquots that have been used to obtain the values in the figure. |
| 4. In Figure 6c, it is mentioned in line 388 of the text that the OD results of the filtered feldspar and quartz are very similar. However, comparing Figure 5 and Figure 6c separately is not intuitive. It may be worth considering including the OD of quartz in Figure 6c for comparison. | We shall add the quartz OD values into Figure 6c as recommended. |
| 5. In Figure 7, what does each point represent? Are they the age results of each sample? More detailed explanations should be provided in the caption. | Figure 7 is intended to show the effects of the filtering methods by comparing the De obtained from each dataset. The points of Figure 7.a. represent the ratio of the samples, and the points of Figure 7.b. demonstrate the ratio / depth relationship. We shall add this information to the caption as suggested. |
| 6. Table 3 should include a depth column for easier comparison with the figures. | We shall add the depth column to Table 3. |
| **For lines** | |
| Line 17, "Recently, luminescence… have recently", one of the "recently" should be removed. | We shall remove the latter one. |
| Line 69, the second goal has not been introduced in the previous text. Why is it important to identify changes in disturbances? What is the significance? | The introduction of plaggen agriculture is one of the major aspects reflecting the increasing land-use intensity during the Medieval Ages in the northwestern Europe. We believe that demonstrating the changes in soil-mixing intensity using luminescence dating techniques can provide a basis for the quantitative estimation of the intensification of land-use by the adoption of plaggen agriculture. We will include this in the paragraph before the research goals while revising the introduction. |
| Line 110, "They conclude that-", who does the "They" refer to? | We shall revise the sentence to "The research by van Mourik et al. (2011) conclude that-", for clarification. |
| Line 240, the sample ID does not match that in the Table 1. | We will correct the sample number accordingly (NCL-1117023 -> NCL-1117123 / NCL-1117029 -> NCL-1117129). |
| Line 265, it is good to consider the influence of fading on the ratio. However, is the ratio of Pirir290 really applicable to Pirir175? Is it possible to obtain a reference value by fitting data from published measurements of IR50 and pirir175 results taken simultaneously? | We adopted this threshold following Buylaert et al. (2013), even though we are using different pIRIR signals than the original publication by those authors. We acknowledge that the 90% threshold is indeed arbitrary; we would expect the ratio to depend on the age of the samples and the fading rate of the signals used for the mineral extract that is measured. We like the suggestion that the threshold could be obtained from a fit between published IRSL50 and pIRIR175 data. |

| | However, given the dependency on sample age and provenance, we argue that a ratio based on published information would be equally arbitrary. Moreover, there is not a lot of published pIRIR175 data for well-bleached samples that would allow comparison of results at single-grain level. |
|---|---|
| Line 268, "To determine the ages of samples", what specific ages are being referred to? If it refers to the poorly-bleached sample, it is understandable to use MAM to determine the depositional age. However, since the filtered pIRIR ages are already from well-bleached grains, why not use CAM to derive the depositional age? Actually, in your context, it doesn't seem like MAM is being used to obtain the conventional "depositional age", correct? So, this should be explained in more detail. | You are definitely correct that the ages referred to here are not the "depositional age". The BsMAM ages rather reflect the latest temporal period that the grains have been surfaced by soil reworking process. We will take your comment into account and clarify this in the revision by adding a conceptual diagram of how bleaching works in soils under effects of bioturbation and agricultural activities. For using CAM, we agree that this would be effective for the samples collected from the plaggen deposits. For these samples, we would like to emphasize that BsMAM and CAM provide identical results provided that the correct sigma_b is used (Chamberlain et al., 2018). However, for the samples collected from deeper horizons, which have been exposed to prolonged (i.e. less intensive) soil reworking processes, we believe the CAM is less likely to provide the depositional age as well. Also, the main focus of this research is centered on the soil mixing rather than deposition, therefore we mainly utilize BsMAM ages. |
| Line 291, why is the sigmab input for quartz determined as 0.15±0.04? | We have applied BsMAM to the OD obtained from CAM results of quartz measurements, as introduced by Chamberlain et al. (2018). We shall add this information to the revised manuscript. |
| Line 373, why is it at most an overestimate of 30%? Isn't there unfiltered/filtered ratio over 1.5? | We intended to mention that the average of the overestimate was about 30%. We shall revise the sentence to make it clear. |
| Line 473, I now understand that the high proportion of poorly bleached grains in the plaggen layer can be attributed to intensive cultivation activities, as you have clearly explained. However, why can we infer that the sedimentation rate also increased during the same time? | We thought that the increase in sedimentation rate was visible in Figure 8, but it may not be as clear as we thought. We will provide additional information on the sedimentation rate calculated by depth / luminescence ages. |
| Line 488, I am not arguing against the idea of using MaxAM to estimate the depositional age. However, it should be noted that bioturbation not only introduces younger grains but can also bring older grains from lower layers. Therefore, the use of MaxAM cannot completely eliminate the influence of bioturbation. | We agree that bioturbation may indeed introduce older grains from deeper deposits. However, in this specific context this is less likely as evidence from the broader region suggests that there is quite a thick layer of coversand that was deposited in a short period of time. This implies that bioturbation mixes sediments of similar depositional age. Also, it should be kept in mind that bioturbation is performed by bioturbating agents (e.g. earthworms) and they need a life supporting soil-food-web that is not present in the underlying Cg horizon due i) ground-water fluctuation, ii) no humus accumulation in purified quartz sand. We will discuss and clarify in the revised manuscript. |
| Line 496, the expression of this age is somewhat confusing. I suppose it should be "900-1000 years ago"? The same issue applies to Line 503. Please check the consistency of age expression throughout the article, abstract, and discussion sections. | We agree that the expression of age can cause confusion and consistency is important. We will check the full manuscript on the consistency of age expressions. |
| Line 517, "The ratio of DeIRSL~" at single grain scale. As you have mentioned that the ratio has already been applied in single-aliquot. | We will add "at single-grain scale" as commented. |

| | |
|---|---|
| Line 519, Single-grain feldspar pIRIR yields similar ages as single-aliquot quartz OSL ages when~. | We shall revise the sentence as advised. |
| Lastly, I am interested to know if the authors have checked the variations in the proportions of zero-age grains throughout the profile. | This dataset contained very few zero-age grains (within 1-sigma error), and these were only found in the topmost sample (NCL-1117134). These results suggest that modern mixing is restricted to the upper layer, and that samples were not contaminated with modern material during sampling or processing. |

---

## Author Comment (AC2)

| Comments | Responses |
|---|---|
| L147 - are you sure that veneered is the right word? It's beautiful! But it feels not quite right. Maybe 'covered' is better? | We agree that the suggested 'covered' would be more straightforward and clear to the readers. We will revise as suggested. |
| L160 - I'd like to read more about the location of the pit inside the B. site. It may be important for your interpretation whether the site is right in the middle, on the edge, close to the river, etc. | The pit was dug on the middle of a field plot, which is located on the outer border of a larger area covered with plaggic anthrosols, as shown in Figure 2. The field plot is adjacent to a stream valley, including a tributary of the Dinkel. We shall include this information as commented. |
| L163 - Be clear about which soil classification system you're using also in the text. You're using Dutch classifiers for your horizons (Aap for instance). The FAO system, which I think you should adopt, has different meaning for the lowercase a than the dutch system (see page 72 https://www.fao.org/3/a0541e/a0541e.pdf). Since most readers will know the FAO system but not the dutch system, please either switch to FAO (recommended) or spend some space on explaining what Aap and other codes mean to non-Dutch speakers. | We will provide more detailed description on the suffixes for the readers. We do agree that adopting the FAO system would be most convenient for the readers, but it would be difficult to make direct transition from the Dutch system to the FAO system at this stage. |
| L181 I'd appreciate a picture of the tubes inside the pit, with a scale-ribbon. Reference your figure 5 here, or duplicate its right half as your new figure 3. That's less abstract than reading the depths from Table 1. | We agree that adding the figure would be useful. We will present the picture along with Table 1, making it Figure 3. |
| L210 You mentioned using the Wageningen lab for pre work. Were the readers also there, or in Koeln? | All of the luminescence sample preparations and measurements (incl gamma spectrometry) were performed in Wageningen. We will make it more clear during revision. |
| L313 It seems the test results are well documented here in the results. Well done, and appropriate given your first objective. They are results here (otherwise, they might have fitted in your methods). | Thank you. |
| Figure 5: please extend also the top dashed line to the left of the figure | We will revise the figure as commented. |
| L327 can you argue with numbers that there is no clear trend? I see a weak trend that is the opposite of your expectation, which would be interesting/require explanation/speculation if it were significant | We will apply linear regression to the results and see whether there is indeed a trend. We will also add a brief discussion about it in Section 5.1. |
| In Figure 8 and Table 2, can you add a column with years in CE? There may be a standard that calls for years before 2017, but it feels easier to read for a wider audience with years CE as a secondary x-axis or extra column. | We will add a column with years in CE as recommended. |

---

## Author Comment (AC3)

| Comments | Responses |
|---|---|
| **Major comments** | |
| 1) From the application of a Minimum Age Model (here bootstrapped MAM) to both quartz OSL and feldspar pIRIR signals the reader would infer that both the feldspar pIRIR and the quartz OSL signals are partially reset or show (high) dose tails (possibly due to mixing). However, at least 3 samples (NCL-11171 28,29 &30) have an quartz OSL over-dispersion (OD) of ~15% (see Fig. 5) and this is identical to the input OD of a well-bleached sample for BsMAM modeling. So, one would consider these samples as well-bleached for quartz OSL. Do the BsMAM and the CAM (or weighted or unweighted means if the authors prefer) give the same answer as the BsMAM for these samples? If the BsMAM works for well-bleached material both models should return identical results and the authors should demonstrate this. The CAM results should be listed and compared with BsMAM. Actually, the authors themselves allude on samples with well-bleached quartz OSL characteristics (see lines 394-395); please show the quartz OSL De distributions for all samples in Supp Info. | You are correct in that the three samples provide the same CAM and BsMAM provide the same results (agreeing with 1-sigma error) as we used the systematic approach of assigning a meaningful sigma_b value to our analyses outlined by Chamberlain et al (2018) . We shall provide the CAM results as commented, and demonstrate that for well-bleached samples results for CAM and BsMAM are in agreement. Also, we would like to point out that signal bleaching in our setting is most likely function of soil mixing intensity. The grains get surfaced by either natural bioturbation or anthropogenic ploughing. Thus, the dose tails mentioned by the reviewer are most likely a result of incomplete mixing rather than due to mixing as suggested by the reviewer. We will include a conceptual diagram to clarify our points on how soil mixing affects bleaching and dose distributions. |
| Would it be possible to discuss the average IR50 results in this paper too? If the IR50 signal is sufficiently reset which is definitely possible for the three samples mentioned above, these samples are likely to give IR50 ages smaller than quartz OSL (because of fading). The samples that are less-well bleached for both IR50 and pIRIR180 signals will tend to give ages equal to or larger than quartz OSL. I miss a discussion at the level of the average behaviour (CAM, weighted mean) in this manuscript. | We will include the analysis on IRSL signals as suggested. We have a similar expectation that IR50 ages will provide younger ages when compared to pIRIR175 ages. For well-bleached samples, as mentioned above, the BsMAM age from pIRIR175 provide same results with the CAM age from quartz OSL. We will report the results in the revised manuscript. |
| This leads me to the proposed research question: 1) How can well-bleached grains be identified for feldspar single-grain pIRIR dating? In my view, to answer this question one needs some form of independent age control. I cannot find that in this paper, especially because the authors believe that the quartz OSL ages should also be inferred from a MAM approach. The best option in the case one does not have independent age control, would be to use a well-bleached, unmixed quartz age (from CAM) and compare with the MAM age of feldspar pIRIR (filtered or unfiltered). | We agree that a truly independent age control would be ideal to answer the proposed research question. However, dating plaggic soil through other means is problematic. However, we have shown that quartz OSL for three samples are well bleached (low overdispersion and BsMAM in agreement with CAM results). We argue that the results on these samples are highly robust, and provide good age control to test our feldspar single grain dating. |
| 2) I cannot seem to find the radionuclide concentrations, used water contents and the total dose rates in the paper. These data are crucial to calculate luminescence ages and should be tabulated. | We shall provide the data essential to calculate the dose rate in the revised version. |
| **Minor comments** | |
| Suppl Mat A.1 (Table): suggest to change cutheat to preheat. Cutheat refers to immediate cooling after reaching temperature but test dose preheat here has duration of 10s. | We will change the term in the revised version. |
| Line 17: humans, remove second recently, | We will remove the second 'recently'. |
| Line 36: has created | Will be corrected as commented. |
| Line 39: factor in the creation of anthrosols? | Will be revised as commented. |
| Line 56 (caption): at Braakmankamp | Will be corrected as commented. |
| Line 79: remove full stop after question mark | Full stop will be removed in the revised version. |

| | |
|---|---|
| Line 102: northern | Will be corrected as commented. |
| Line 145: place at a site | Will be corrected as commented. |
| Line 159 (caption): in areas with coversand | Will be corrected as commented. |
| Line 162: At all depths | Will be corrected as commented. |
| Line 169: gleiing | Will be corrected as commented. |
| The K-feldspar grains were not etched. Did you take into account an external alpha contribution? If so, how large is it? | The external alpha contribution was taken into account with the assumption of 0.05±0.025 Gy/ka. |
| Line 246-247: These contrasting effects, von Suchodoletz | Will be corrected as commented. |
| Line 252: Poolton et al. looked at elevated temperature IRSL but not post-IR elevated temperature IRSL? Please check, if not pIRIR, then remove ref. | You are correct; we shall remove the reference. |
| Lines 285-286: not logical after previous sentence in which it is stated that TT is very small or negligible (at least I cannot see a trend). There is more scatter in the results but this is not necessarily due to thermal transfer? Can also be sensitivity changes not full accounted for by test dose? Please rephrase. | We agree that there is a larger scatter, rather than a small TT, and will rephrase this. |
| Line 324: majority of the samples | Will be corrected as commented. |
| Line 343: Fig. 7a | Will be corrected as commented. |
| Line 346: remove second full stop | Will be corrected as commented. |

---

## Author Response (AR1)

<Reviewer 1>

Dear reviewer,

We would like to thank you for your positive and constructive comments. We have incorporated the large majority of your suggestions to further improve the quality of the manuscript.

One comment that gained our attention centers on section divisions, suggesting that the second section might be merged with the first section. We have discussed with the author team and concluded that it may be better to keep the introduction section relatively short and direct to highlight the research questions and objectives. In contrast, much more detail is needed in the subsequent section to help members of both the archeological and luminescence communities to grasp the study and its technical and geographic aspects. However, we agreed that there was space for improvement in the introduction section, and we have revised it.

Another important comment relates to the 90% criteria for the filtering method (De ratio IRSL50 / pIRIR175). We adopted this threshold following Buylaert et al. (2013), even though we are using a different stimulation temperature of the pIRIR signal than the original publication by those authors. We acknowledge in the manuscript that the 90% threshold is indeed arbitrary; we would expect the ratio to depend on the age of the samples and the fading rate of the signals used for the mineral extract that is measured. We like the suggestion that the threshold could be obtained from a fit between published IRSL50 and pIRIR175 data. However, given the dependency on sample age and provenance, we argue that a ratio based on published information would be equally arbitrary. Moreover, there is not a lot of published pIRIR175 data for well-bleached samples that would allow a comparison of results at a single-grain level.

For our detailed response to the rest of the comments, please check the list.

| Comments | Response |
|---|---|
| **For sections** | |
| 1. The abstract is a little bit cumbersome, consider deleting the second and third sentences. | We have revised the abstract and replaced the mentioned sentences. |
| 2. The second section might be merged into the first section, placed just before presenting the scientific questions and research objectives. | We prefer to keep the introduction section relatively short, to help highlight the research questions and objectives. The subsequent 'dating plaggen soils' section is longer and more detailed because it provides the background needed for readers of different expertise (archeological and luminescence communities) to grasp the study and its technical aspects. |
| **For figures** | |
| 1. Figure 2 should be labeled with latitude and longitude. | We added the grids and coordinates to the figure, which is now Figure 3. |
| 2. The title of Figure 3 (b)'s z-coordinate should be changed to "Dose recovery ratio". | We changed the label of the z-coordinate as recommended, which can now be seen in Figure 4 (b). |
| 3. In the caption of Figure 4, it should be specified that the average value is calculated from how many results. | We added the number of grains / aliquots that have been used to obtain the values in the figure and the caption for which is now Figure 5. |
| 4. In Figure 6c, it is mentioned in line 388 of the text that the OD results of the filtered feldspar and quartz are very similar. However, comparing Figure 5 and Figure 6c separately is not intuitive. It may be worth considering including the OD of quartz in Figure 6c for comparison. | We added the quartz OD values to Figure 6c as recommended. |
| 5. In Figure 7, what does each point represent? Are they the age results of each sample? More detailed explanations should be provided in the caption. | Figure 7 is intended to show the effects of the filtering methods by comparing the De obtained from each dataset. The points of Figure 7.a. represent the ratio of the samples and the points of Figure 7.b. demonstrate the ratio / depth relationship. We added this information to the caption as suggested. |
| 6. Table 3 should include a depth column for easier comparison with the figures. | We added the depth column to Table 3. |
| **For lines** | |
| Line 17, "Recently, luminescence… have recently", one of the "recently" should be removed. | We have removed the latter one. (Line 16) |
| Line 69, the second goal has not been introduced in the previous text. Why is it important to identify changes in disturbances? What is the significance? | The introduction of plaggen agriculture is one of the major aspects reflecting the increasing land-use intensity during the Medieval Ages in northwestern Europe. We believe that demonstrating the changes in soil-mixing intensity using luminescence dating techniques can provide a basis for the quantitative estimation of the intensification of land-use by the adoption of plaggen agriculture. We included this in the paragraph before the research goals while revising the introduction. (Lines 55-59) |
| Line 110, "They conclude that-", who does the "They" refer to? | We revised the sentence to "The research by van Mourik et al. (2011) conclude that-", for clarification. (Line 120) |
| Line 240, the sample ID does not match that in the Table 1. | We corrected the sample number accordingly (NCL-1117023 -> NCL-1117123 / NCL-1117029 -> NCL-1117129). (Line 258) |
| Line 265, it is good to consider the influence of fading on the ratio. However, is the ratio of Pirir290 really applicable to Pirir175? Is it possible to obtain a reference value by fitting data from published measurements of IR50 and pirir175 results taken simultaneously? | We adopted this threshold following Buylaert et al. (2013), even though we are using different pIRIR signals than the original publication by those authors. We acknowledge that the 90% threshold is indeed arbitrary; we would expect the ratio to depend on the age of the samples and the fading rate of the signals |

| | used for the mineral extract that is measured. We like the suggestion that the threshold could be obtained from a fit between published IRSL50 and pIRIR175 data. However, given the dependency on sample age and provenance, we argue that a ratio based on published information would be equally arbitrary. Moreover, there is not a lot of published pIRIR175 data for well-bleached samples that would allow a comparison of results at a single-grain level. |
|---|---|
| Line 268, "To determine the ages of samples", what specific ages are being referred to? If it refers to the poorly-bleached sample, it is understandable to use MAM to determine the depositional age. However, since the filtered pIRIR ages are already from well-bleached grains, why not use CAM to derive the depositional age? Actually, in your context, it doesn't seem like MAM is being used to obtain the conventional "depositional age", correct? So, this should be explained in more detail. | You are definitely correct that the ages referred to here are not the "depositional age". The BsMAM ages rather reflect the latest temporal period that the grains have been surfaced by the soil reworking process. We have taken your comment into account and clarified this in the revision by adding a conceptual diagram of how bleaching works in soils under the effects of bioturbation and agricultural activities (Figure 2). For using CAM, we agree that this would be effective for the samples collected from the plaggen deposits. For these samples, we would like to emphasize that BsMAM and CAM provide identical results provided that the correct sigma_b is used (Chamberlain et al., 2018). However, for the samples collected from deeper horizons, which have been exposed to prolonged (i.e. less intensive) soil reworking processes, we believe the CAM is less likely to provide the depositional age as well. Also, the main focus of this research is centered on soil mixing rather than deposition, therefore we mainly utilize BsMAM ages. |
| Line 291, why is the sigmab input for quartz determined as 0.15±0.04? | We have applied BsMAM to the OD obtained from CAM results of quartz measurements, as introduced by Chamberlain et al. (2018). We added this information to the revised manuscript. (Lines 312-314) |
| Line 373, why is it at most an overestimate of 30%? Isn't there unfiltered/filtered ratio over 1.5? | We intended to mention that the average of the overestimate was about 30%. We revised the sentence to make it clear. (Line 401) |
| Line 473, I now understand that the high proportion of poorly bleached grains in the plaggen layer can be attributed to intensive cultivation activities, as you have clearly explained. However, why can we infer that the sedimentation rate also increased during the same time? | We thought that the increase in sedimentation rate was visible in Figure 8, but it may not be as clear as we thought. We provided additional information on the sedimentation rate calculated by depth / luminescence ages. (Section 5.3, Last Paragraph) |
| Line 488, I am not arguing against the idea of using MaxAM to estimate the depositional age. However, it should be noted that bioturbation not only introduces younger grains but can also bring older grains from lower layers. Therefore, the use of MaxAM cannot completely eliminate the influence of bioturbation. | We agree that bioturbation may indeed introduce older grains from deeper deposits. However, in this specific context, this is less likely as evidence from the broader region suggests that there is quite a thick layer of coversand that was deposited in a short period of time. This implies that bioturbation mixes sediments of similar depositional age. Also, it should be kept in mind that bioturbation is performed by bioturbating agents (e.g. earthworms) and they need a life-supporting soil-food-web that is not present in the underlying Cg horizon due to i) ground-water fluctuation, ii) no humus accumulation in purified quartz sand. We discussed this in more detail in the revised manuscript. (Section 5.4, Second Paragraph) |
| Line 496, the expression of this age is somewhat confusing. I suppose it should be "900-1000 years | We agree that the expression of age can cause confusion and consistency is important. We checked |

| | |
|---|---|
| ago"? The same issue applies to Line 503. Please check the consistency of age expression throughout the article, abstract, and discussion sections. | the manuscript on the consistency of age expressions and made revisions accordingly. |
| Line 517, "The ratio of DeIRSL~" at single grain scale. As you have mentioned that the ratio has already been applied in single-aliquot. | We added "for single-grains" for clarification. (Line 579) |
| Line 519, Single-grain feldspar pIRIR yields similar ages as single-aliquot quartz OSL ages when~. | We revised the sentence as advised. (Line 583) |
| Lastly, I am interested to know if the authors have checked the variations in the proportions of zero-age grains throughout the profile. | This dataset contained very few zero-age grains (within 1-sigma error), and these were only found in the topmost sample (NCL-1117134). These results suggest that modern mixing is restricted to the upper layer and that samples were not contaminated with modern material during sampling or processing. |

<Reviewer 2>

Thank you for your positive and constructive comments. We found them very useful to improve our manuscript. We tried to incorporate most of your suggestions.

One comment that we discussed over is whether to adopt the FAO system for the descriptions of the soil horizons. Although we do agree that using the FAO system would be most familiar for the majority of the readers, it seems that it would be difficult to make a direct transition from the Dutch system to the FAO system. As an alternative, we provided detailed descriptions for the suffixes used for the descriptions of the soil horizons, which would be sufficient enough for the readers to get a good understanding of the soil profile.

For our response to the rest of the comments, please check the list presented below.

| Comments | Responses |
|---|---|
| L147 - are you sure that veneered is the right word? It's beautiful! But it feels not quite right. Maybe 'covered' is better? | We agree that the suggested 'covered' would be more straightforward and clear to the readers. We replaced it as suggested. (Line 159) |
| L160 - I'd like to read more about the location of the pit inside the B. site. It may be important for your interpretation whether the site is right in the middle, on the edge, close to the river, etc. | The pit was dug in the middle of a field plot, which is located on the outer border of a larger area covered with plaggic anthrosols, as shown in Figure 2. The field plot is adjacent to a stream valley, including a tributary of the Dinkel. We included this information in the revised manuscript as commented. (Lines 173-175) |
| L163 - Be clear about which soil classification system you're using also in the text. You're using Dutch classifiers for your horizons (Aap for instance). The FAO system, which I think you should adopt, has different meaning for the lowercase a than the dutch system (see page 72 https://www.fao.org/3/a0541e/a0541e.pdf). Since most readers will know the FAO system but not the dutch system, please either switch to FAO (recommended) or spend some space on explaining what Aap and other codes mean to non-Dutch speakers. | We do agree that adopting the FAO system would be most convenient for the readers, but it would be difficult to make a direct transition from the Dutch system to the FAO system at this stage. Therefore, we provided a more detailed description of the suffixes for the readers. (Table 1) |
| L181 I'd appreciate a picture of the tubes inside the pit, with a scale-ribbon. Reference your figure 5 here, or duplicate its right half as your new figure 3. That's less abstract than reading the depths from Table 1. | We agree that adding the figure would be useful. We presented the picture in Table 1. |
| L210 You mentioned using the Wageningen lab for pre work. Were the readers also there, or in Koeln? | All of the luminescence sample preparations and measurements (including gamma spectrometry) were performed in Wageningen. We revised it to make it more clear in the revised manuscript. (Lines 204, 228-229) |
| L313 It seems the test results are well documented here in the results. Well done, and appropriate given your first objective. They are results here (otherwise, they might have fitted in your methods). | Thank you. |
| Figure 5: please extend also the top dashed line to the left of the figure | The figure has been removed during the revision process. |
| L327 can you argue with numbers that there is no clear trend? I see a weak trend that is the opposite of your expectation, which would be interesting/require explanation/speculation if it were significant | We applied linear regression to the results and demonstrated that there is indeed an opposite trend. (Figure 6.b) We also added a brief discussion about it in Section 5.1. (Lines 393-399) |
| In Figure 8 and Table 2, can you add a column with years in CE? There may be a standard that calls for years before 2017, but it feels easier to read for a wider audience with years CE as a secondary x-axis or extra column. | We added years in CE below the ages expressed in a. (Table 2) |

<Reviewer 3>

Dear reviewer,

Thank you for your constructive comments on our paper. We found them very useful to improve our manuscript and are planning to adopt most of them. We agree that the elements that you mentioned in the major comments are essential to understand the dataset and to allow others to reproduce the research. Thanks also for the careful proofreading.

You mentioned that it would be important to show that BsMAM works for well-bleached materials by demonstrating that the two models provide the same results. You are indeed correct that the CAM and the BsMAM results are identical (agreeing with 1-sigma error) for the samples that can be considered as 'well-bleached', or 'completely mixed' by bioturbation or ploughing. We provided the CAM results and demonstrated that well-bleached samples provide similar results for both CAM and BsMAM in the revised manuscript.

We unfortunately do not have an independent age control for this research. We agree that the solution that you provided, using the CAM ages of the well-bleached samples (NCL-1117128 ~ 130), would be the best alternative in this case. The CAM results of quartz OSL and BsMAM results of feldspar pIRIR agree for two of the three samples (NCL-1117128 and NCL-1117130), but there is an underestimation by feldspar pIRIR in NCL-1117129. Despite one disagreement, we think two samples having agreeing ages from both single-aliquot quartz and single-grain feldspar still provide support for our arguments.

For our response to the rest of the comments, please check the list presented below.

| Comments | Responses |
|---|---|
| **Major comments** | |
| 1) From the application of a Minimum Age Model (here bootstrapped MAM) to both quartz OSL and feldspar pIRIR signals the reader would infer that both the feldspar pIRIR and the quartz OSL signals are partially reset or show (high) dose tails (possibly due to mixing). However, at least 3 samples (NCL-11171 28,29 &30) have an quartz OSL over-dispersion (OD) of ~15% (see Fig. 5) and this is identical to the input OD of a well-bleached sample for BsMAM modeling. So, one would consider these samples as well-bleached for quartz OSL. Do the BsMAM and the CAM (or weighted or unweighted means if the authors prefer) give the same answer as the BsMAM for these samples? If the BsMAM works for well-bleached material both models should return identical results and the authors should demonstrate this. The CAM results should be listed and compared with BsMAM.

Actually, the authors themselves allude on samples with well-bleached quartz OSL characteristics (see lines 394-395); please show the quartz OSL De distributions for all samples in Supp Info. | You are correct in that the three samples provide the same CAM and BsMAM provide the same results (agreeing with 1-sigma error) as we used the systematic approach of assigning a meaningful sigma_b value to our analyses outlined by Chamberlain et al (2018). We provided the CAM results as commented, and demonstrated that for well-bleached samples results for CAM and BsMAM are in agreement. (Table 4)

Also, we would like to point out that signal bleaching in our setting is most likely a function of soil mixing intensity. The grains get surfaced by either natural bioturbation or anthropogenic ploughing. Thus, the dose tails mentioned by the reviewer are most likely a result of incomplete mixing rather than due to mixing as suggested by the reviewer. We included a conceptual diagram to clarify our points on how soil mixing affects bleaching and dose distributions. (Figure 2) |
| Would it be possible to discuss the average IR50 results in this paper too? If the IR50 signal is sufficiently reset which is definitely possible for the three samples mentioned above, these samples are likely to give IR50 ages smaller than quartz OSL (because of fading). The samples that are less-well bleached for both IR50 and pIRIR180 signals will tend to give ages equal to or larger than quartz OSL. I miss a discussion at the level of the average behaviour (CAM, weighted mean) in this manuscript. | We included the analysis on IRSL signals as suggested. The IR50 ages indeed provided younger ages when compared to pIRIR175 ages. (Lines 465-470) For well-bleached samples, as mentioned above, the BsMAM age from pIRIR175 provides the same results as the CAM age from quartz OSL. We reported the results in the revised manuscript and also included the full result in the supplementary material in the form of radial plots. |
| This leads me to the proposed research question: 1) How can well-bleached grains be identified for feldspar single-grain pIRIR dating? In my view, to answer this question one needs some form of independent age control. I cannot find that in this paper, especially because the authors believe that the quartz OSL ages should also be inferred from a MAM approach. The best option in the case one does not have independent age control, would be to use a well-bleached, unmixed quartz age (from CAM) and compare with the MAM age of feldspar pIRIR (filtered or unfiltered). | We agree that a truly independent age control would be ideal to answer the proposed research question. However, dating plaggic soil through other means is problematic. However, we have shown that quartz OSL signals for three samples are well bleached (low overdispersion and BsMAM in agreement with CAM results). We argue that the results on these samples are highly robust, and provide good age control to test our feldspar single grain dating. |
| 2) I cannot seem to find the radionuclide concentrations, used water contents and the total dose rates in the paper. These data are crucial to calculate luminescence ages and should be tabulated. | We provided the data essential to calculate the dose rate in the revised supplementary material. |
| **Minor comments** | |
| Suppl Mat A.1 (Table): suggest to change cutheat to preheat. Cutheat refers to immediate cooling after reaching temperature but test dose preheat here has duration of 10s. | We changed the term in the revised version. (Supplementary Material B.2) |
| Line 17: humans, remove second recently, | We removed the second 'recently'. (Line 16) |
| Line 36: has created | Sentence removed during revision. |
| Line 39: factor in the creation of anthrosols? | Sentence removed during revision. |

| | |
|---|---|
| Line 56 (caption): at Braakmankamp | Corrected as commented. (Line 49) |
| Line 79: remove full stop after question mark | Full stop removed in the revised version. (Line 77) |
| Line 102: northern | Corrected as commented. (Line 99) |
| Line 145: place at a site | Corrected as commented. (Line 156) |
| Line 159 (caption): in areas with coversand | Corrected as commented. (Line 171) |
| Line 162: At all depths | Corrected as commented. (Line 175) |
| Line 169: gleiing | Changed from "gleiing phenomena are" to "gleiing is". (Line 183) |
| The K-feldspar grains were not etched. Did you take into account an external alpha contribution? If so, how large is it? | The external alpha contribution was taken into account with the assumption of 0.05±0.025 Gy/ka. We provided the information in the revised manuscript. (Lines 220-221) |
| Line 246-247: These contrasting effects, von Suchodoletz | Corrected as commented. (Line 265) |
| Line 252: Poolton et al. looked at elevated temperature IRSL but not post-IR elevated temperature IRSL? Please check, if not pIRIR, then remove ref. | You are correct; we removed the reference. |
| Lines 285-286: not logical after previous sentence in which it is stated that TT is very small or negligible (at least I cannot see a trend). There is more scatter in the results but this is not necessarily due to thermal transfer? Can also be sensitivity changes not full accounted for by test dose? Please rephrase. | We agree that there is a larger scatter, rather than a small TT, and we have rephrased this. (Lines 307-310) |
| Line 324: majority of the samples | We revised to "majority of the grains". (Line 341) |
| Line 343: Fig. 7a | Corrected as commented. (Line 364) |
| Line 346: remove second full stop | Corrected as commented. (Line 366) |